# On Feature Learning in the Presence of Spurious Correlations

**Pavel Izmailov**[*]   **Polina Kirichenko**[*]   **Nate Gruver**[*]   **Andrew Gordon Wilson**
New York University

## Abstract

Deep classifiers are known to rely on spurious features — patterns which are correlated with the target on the training data but not inherently relevant to the learning problem, such as the image backgrounds when classifying the foregrounds. In this paper we evaluate the amount of information about the core (non-spurious) features that can be decoded from the representations learned by standard empirical risk minimization (ERM) and specialized group robustness training. Following recent work on Deep Feature Reweighting (DFR), we evaluate the feature representations by re-training the last layer of the model on a held-out set where the spurious correlation is broken. On multiple vision and NLP problems, we show that the features learned by simple ERM are highly competitive with the features learned by specialized group robustness methods targeted at reducing the effect of spurious correlations. Moreover, we show that the quality of learned feature representations is greatly affected by the design decisions beyond the training method, such as the model architecture and pre-training strategy. On the other hand, we find that strong regularization is not necessary for learning high-quality feature representations. Finally, using insights from our analysis, we significantly improve upon the best results reported in the literature on the popular Waterbirds, CelebA hair color prediction and WILDS-FMOW problems, achieving 97%, 92% and 50% worst-group accuracies, respectively.

## 1   Introduction

In classification problems, a feature is *spurious* if it is predictive of the label without being causally related to it. Models that exploit the predictive power of spurious features can achieve strong average performance on training and in-distribution test data but often perform poorly on sub-groups of the data where the spurious correlation does not hold [18]. For example, neural networks trained on ImageNet are known to rely on backgrounds [93] or texture [19], which are often correlated with labels without being causally significant. Similarly in natural language processing, models often rely on specific words and syntactic heuristics when predicting the sentiment of a sentence or the relationship between a pair of sentences [56, 22]. In extreme cases, neural networks completely ignore task-relevant *core* features and only use spurious features in their predictions [97, 79], achieving zero accuracy on the subgroups of the data where the spurious correlation does not hold.

In recent work, Kirichenko et al. [40] showed that, surprisingly, standard Empirical Risk Minimization (ERM) learns a high-quality representation of the core features on datasets with spurious correlations, even when the model primarily relies on spurious features to make predictions. Moreover, they showed that it is often possible to recover state-of-the-art performance on benchmark problems by simply retraining the last layer of the model on a small held-out dataset where the spurious correlation does not hold. This procedure is called Deep Feature Reweighting (DFR).

---

[*]Equal contribution.

36th Conference on Neural Information Processing Systems (NeurIPS 2022).

In this paper, we provide an in-depth study of the factors that affect the *quality of learned representations* in the presence of spurious correlations: how accurately can we decode the core features from the learned representations. Following Kirichenko et al. [40], we break the problem of training a robust classifier into two tasks: extracting feature representations and training a linear classifier on these features. In order to study the feature learning in isolation, we use the DFR procedure to learn an optimal linear classifier on the feature representations, and evaluate the features learned with different training methods, neural network architectures, and hyper-parameters.

First, on a range of problems with spurious correlations we show that while specialized group robustness methods such as group distributionally robust optimization (group DRO) [76] can significantly outperform the standard ERM training, the quality of the features learned by ERM is highly competitive: by applying the DFR procedure to the features learned by ERM and group DRO we achieve similar performance. Furthermore, we show that the performance improvements of group DRO are largely explained by the better weighting of the learned features in the last classification layer, and not by learning a better representation of the core features. This observation has high practical significance, as the problem of training the last layer of the model is much simpler both conceptually and computationally than training the full model to avoid spurious correlations [40].

Next, focusing on the ERM training, we explore the effect of model class, pretraining strategy and regularization on feature learning. We find a linear dependence between the in-distribution accuracy of the model and the worst group accuracy after applying DFR, meaning that on natural datasets good generalization typically implies good feature learning, even in the presence of spurious features. Further, we show that the pre-training strategy has a very significant effect on the quality of the learned features, while strong regularization does not significantly improve the feature representations on most benchmarks.

Finally, by finetuning a state-of-the-art ConvNext model [52], we significantly outperform the best reported results on the popular Waterbirds [76], CelebA hair color and WILDS FMOW [43] spurious correlation benchmarks, using only simple ERM training followed by DFR.

Our code is available at `github.com/izmailovpavel/spurious_feature_learning`.

## 2   Related Work

Numerous works describe how neural networks can rely on spurious correlations in real world problems. In vision, neural networks can learn to rely on an image's background [93, 76, 61], secondary objects [45, 72, 83, 80, 2, 60], object textures [19] and other semantically irrelevant features [9, 49]. Spurious correlations are especially problematic in high-risk domains such as medical imaging, where it was shown that neural networks can use hospital-specific metal tokens [97] or cues of disease treatment [65] rather than symptoms to perform automated diagnosis on chest X-ray images. Spurious features are also extremely prevalent in NLP, where models can achieve good performance on benchmarks without properly solving them, e.g. by using simple syntactic heuristics such as lexical overlap between the two sentences in order to classify the relationship between them [64, 22, 37, 56]. For a comprehensive survey of the area, see Geirhos et al. [19].

Because of the high practical significance of spurious correlations, many *group robustness* methods have been proposed. These methods aim to reduce the reliance of deep learning models on spurious correlations and improve worst group performance. Group DRO [76] is the state-of-the-art group robustness method, which minimizes the worst-group loss instead of the average loss. Other works focus on automatically identifying the minority group examples [50, 62, 13, 100], learning several diverse classifiers that use different features [48, 66, 85] or using partially available group labels [84, 63]. Group subsampling was shown to be a strong baseline for some benchmarks [34, 78].

In this work, we focus on *feature learning* in the presence of spurious correlations. Hermann and Lampinen [31] perform a conceptually similar study, but focusing on synthetic datasets. Similar to their work, we explore how well the different features of the data can be decoded from the features learned by deep neural networks, but on large-scale natural datasets. Hermann et al. [30] explore the feature learning in the context of texture bias [19], finding that data augmentation has a profound effect on the texture bias while architectures and training objectives have a relatively small effect. Ghosal et al. [20] show that Vision Transformer models pretrained on ImageNet22k [44] significantly outperform standard CNN models on several spurious correlation benchmarks.

Lovering et al. [54] explore the factors which affect the extractability of features after pre-training and fine-tuning of NLP models. Kaushik et al. [36] construct counterfactually augmented sentiment analysis and naural language inference datasets (CAD) and show that combining CAD with the original data reduces the reliance on spurious correlations on the corresponding benchmarks. Kaushik et al. [38] explain the efficacy of CAD and show that while adding noise to causal features degrades in-distribution and out-of-distribution performance, adding noise to non-causal features improves robustess. Eisenstein [17] and Veitch et al. [86] formally define and study spurious features in NLP from the perspective of causality.

Kirichenko et al. [40] show that models trained with standard ERM training often learn high-quality representations of the core features, and propose the DFR procedure (see Section 3) which we use extensively in this paper. Related observations have also been reported in other works in the context of spurious correlations [58], domain generalization [73] and long-tail classification [35]. While we build on the observations of Kirichenko et al. [40], our work provides profound new insights and greatly expands on the scope of their work. In particular, we investigate the feature representations learned by methods beyond standard ERM, and the role of model architecture, pre-training, regularization and data augmentation on learning semantic structure. We also extend our analysis beyond the standard spurious correlation benchmarks studied by Kirichenko et al. [40], by considering the challenging real world satellite imaging and chest X-ray datasets.

In an independent and concurrent work, Shi et al. [81] also propose an evaluation framework for out-of-distribution generalization based on last layer retraining, inspired by the observations of Kirichenko et al. [40] and Kang et al. [35]. They focus on the comparison of supervised, self-supervised and unsupervised training methods, providing complementary observations to our work.

## 3   Background

**Preliminaries.**    We consider classification tasks with inputs $x \in \mathcal{X}$ and classes $y \in \mathcal{Y}$. We assume that the data distribution consists of groups $\mathcal{G}$ which are not equally represented in the training data. The distribution of groups can change between the training and test distributions, with majority groups becoming less common or minority groups becoming more common. Because of the imbalance in training data, models trained with ERM often have a gap between average and worst group performance on test. Throughout this paper, we will be studying *worst group accuracy* (*WGA*), i.e. the lowest test accuracy across all the groups $\mathcal{G}$. For most problems considered in this paper, we assume that each data point has an attribute $s \in \mathcal{S}$ which is spuriously correlated with the label $y$, and the groups are defined by a combination of the label and spurious attribute: $\mathcal{G} \in \mathcal{Y} \times \mathcal{S}$. In test distribution we might find that $s$ is no longer correlated with $y$, and thus a model that has learned to rely on the spurious feature $s$ during training will perform poorly at test time. Models that rely on the spurious features will typically achieve poor worst group accuracy, while models that rely on core features will have more uniform accuracies across the groups. In Appendix A, we describe the groups, spurious and core features in the datasets that we use in this paper.

In order to perform controlled experiments, we assume that we have access to the spurious attributes $s$ (or group labels) for training or validation data, which we use for training of group robustness baselines and feature quality evaluation. However, we emphasize that our results on the features learned by ERM hold generally, even when spurious attributes are unknown, as ERM does not use the information about the spurious features: we only use the spurious attributes to perform analysis.

**Deep feature reweighting.**    Suppose we are given a model $m : \mathcal{X} \to \mathcal{C}$, where $\mathcal{X}$ is the input space and $\mathcal{C}$ is the set of classes. Kirichenko et al. [40] assume that the model $m$ consists of a feature extractor (typically, a sequence of convolutional or transformer layers) followed by a classification head (typically, a single linear layer): $m = h \circ e$, where $e : \mathcal{X} \to \mathcal{F}$ is a feature extractor and $h : \mathcal{F} \to \mathcal{C}$ is a classification head. They discard the classification head, and use the feature extractor $e$ to compute the set of embeddings $\hat{\mathcal{D}}_e = \{(e(x_i), y_i)\}_{i=1}^n$ of all the datapoints in the reweighting dataset $\hat{\mathcal{D}}$; the reweighting dataset is used to retrain the last layer of the model, and contains group-balanced data where the spurious correlation does not hold. Finally, they train a logistic regression classifier $l : \mathcal{F} \to \mathcal{C}$ on the dataset $\hat{\mathcal{D}}_e$. For stability, logistic regression models are trained 10 times on different random group-balanced subsets of the reweighting dataset $\hat{\mathcal{D}}$, and the weights of the learned logistic regression models are averaged (see Appendix B of Kirichenko et al. [40] for full details on the DFR procedure). Then, the final model used on new test data is given by $m_l = l \circ e$.

Throughout this paper, we use a group-balanced held-out dataset (subset of the validation dataset where each group has the same number of datapoints) as the reweighting dataset $\hat{\mathcal{D}}$; Kirichenko et al. [40] denote this variation of the method as $\mathrm{DFR}_{\mathrm{Tr}}^{\mathrm{Val}}$.

## 4  Experimental Setup and Evaluation Procedure

In this section, we describe the datasets, models and evaluation procedure that we use throughout the paper.

**Datasets.**    In order to cover a broad range of practical scenarios, we consider four image classification and two text classification problems.

- *Waterbirds* [76] is a binary image classification problem, where the class corresponds to the type of the bird (landbird or waterbird), and the background is spuriously correlated with the class. Namely, most landbirds are shown on land, and most waterbirds are shown over water.

- *CelebA hair color* [51] is a binary image classification problem, where the goal is to predict whether a person shown in the image is blond; the gender of the person serves as a spurious feature, as $94\%$ of the images with the "blond" label depict females.

- WILDS-*FMOW* [12, 43, 77] is a satellite image classification problem, where the classes correspond to one of 62 land use or building types, and the spurious attribute $s$ corresponds to the region (Africa, Americas, Asia, Europe, Oceania or Other; the "Other" region is not used in the evaluation). We note that for the FMOW datasets the groups $\mathcal{G}_s$ are defined by the value of the spurious attribute, and not the combination of the spurious attribute and the class label $\mathcal{G}_{y,s}$, as described in Section 3. Moreover, on FMOW there is also a domain shift: the images for test and validation data (used for last layer retraining) are collected in 2016 and 2017, while the training data is collected before 2016. For more details, please see Appendix A.

- *CXR-14* [89] is a dataset with chest X-ray images for which we focus on a binary classification problem of pneumothorax prediction. Oakden-Rayner et al. [65] showed that there is a hidden stratification in the dataset such that most images from the positive class contain a chest drain, which is a non-causal feature related to treatment of the disease. While for all other benchmarks we report WGA on test data, for this dataset, following prior work [65, 48, 70], we report worst group AUC because of the heavy class imbalance.[1]

- *MultiNLI* [91, 76] is a text classification problem, where the task is to classify the relationship between a given pair of sentences as a contradiction, entailment or neither of them. In this dataset, the presence of negation words (e.g. "never") in the second sentence is spuriously correlated with the "contradiction" class.

- *CivilComments* [8, 43] is a text classification problem, where the goal is to classify whether a given comment is toxic. We follow Idrissi et al. [34] and use the coarse version of the dataset both for training and evaluation, where the spurious attribute is $s = 1$ if the comment mentions at least one of the following categories: male, female, LGBT, black, white, Christian, Muslim, other religion; otherwise, the spurious label is $0$. The presence of the eight categories above is spuriously correlated with the comment being classified as toxic.

The Waterbirds, CelebA, CivilComments and MultiNLI datasets are commonly used to benchmark the performance of group robustness methods [see e.g. 34, 50, 63]. The FMOW and CXR-14 datasets present challenging real-world problems with spurious correlations. In these datasets, the inputs do not resemble natural images from datasets such as ImageNet [75], so models have to learn the relevant features from data to achieve good performance, and cannot simply rely on feature transfer. We provide detailed descriptions of the data and show example datapoints in Appendix A, Figures 6 and 7.

---

[1]We take the minimum out of the two scores on test: AUC for classifying negative class against positive examples with chest drain and AUC for negative class against positive without chest drain.

**Models.**    Following prior work [76, 34] we use a ResNet-50 [26] model pretrained on ImageNet1k [75] on Waterbirds, CelebA and FMOW. For the NLP problems, we use a BERT model [14] pretrained on Book Corpus and English Wikipedia data. On CXR-14, following prior work [e.g. 70, 65, 48] we use a DenseNet-121 model [32] pretrained on ImageNet1k. In Section 6, we provide an extensive study of the effect of architecture and pretraining on the image classification problems, and in Appendix E we perform a similar study on the MultiNLI text classification problem.

**Evaluation strategy.**    We use DFR to evaluate the quality of the learned feature representations, as described in Section 3: we measure how well the core features can be decoded from the learned representations with last layer retraining. In some of the experiments, we also train a classifier to predict the spurious attribute $s$ instead of the class label $y$. Using this classifier, we can evaluate the decodability of the spurious feature from the learned feature representation. We refer to this procedure as $s$-DFR and the corresponding worst group accuracy (in predicting the spurious attribute $s$) as DFR $s$-WGA. Additionally, we evaluate the worst group accuracy and mean[2] accuracy of the base model without applying DFR, which we refer to as *base WGA* and *base accuracy* respectively.

# 5    ERM vs Group Robustness Training

Multiple methods have been proposed for training classifiers which are more robust to spurious correlations, with significant improvements in worst group accuracy compared to standard training. In this section, we use DFR to investigate whether the improvements of group robustness methods are caused by better feature representations or by better weighting of the learned features.

**Methods.**    We consider 4 methods for learning the features. *ERM* or Empirical Risk Minimization is the standard training on the original training data, without any techniques targeted at improving worst group performance. *RWG* reweights the loss on each of the groups according to the size of the group and *RWY* reweights the loss on each class according to the size of the class [34]. *Group DRO* [76] is a state-of-the-art method which uses the group information on the training data to minimize the worst group loss instead of the average loss. Group DRO is often considered as an oracle method or upper-bound on the worst group performance under spurious correlations [50, 13].

On the CXR dataset the group labels are not available on the train data, so we cannot apply RWG or group DRO; on this dataset we compare ERM to RWY. On several datasets, the performance of RWY and RWG methods deteriorates during training. For these datasets, we additionally report the results for the checkpoint obtained with early stopping (RWY-ES and RWG-ES). For group DRO, we report the performance with early stopping on all datasets except for CXR (GDRO-ES). In all cases, early stopping is performed based on the worst-group accuracy on the validation set.

**Hyper-parameters selection.**    We train ERM models, RWG and RWY with the same hyper-parameters shared between all the image datasets (apart from batch size which is set to 32 on Waterbirds, and 100 on the other datasets) and between the natural language datasets. We do not tune the hyper-parameters of these methods for worst group accuracy. For group DRO, we run a grid search over the values of the generalization adjustment $C$, weight decay and learning rate hyper-parameters, and select the best combination according to the worst-group accuracy on validation data with early stopping. For details, please see Appendix B.

## 5.1    Results

We compare feature learning methods on all datasets in Figure 1. As expected, the worst group accuracy of group robustness methods is significantly better than the ERM worst group accuracy on most datasets. For example, on Waterbirds, ERM only gets $68.8\%$ WGA, while group DRO with early stopping gets $90.6\%$. However, after applying DFR the performance of ERM and group DRO is very close, with a slight advantage for ERM ($91.1\%$ for ERM and $90\%$ for group DRO), and similar observations hold on all datasets.

The results for RWG and RWY are analogous. Namely, when combined with early stopping, these methods outperform ERM on base model performance. Once we apply DFR, however, the gap in

---

[2]Following Sagawa et al. [76] and Kirichenko et al. [40], we evaluate the mean accuracy according to the group distribution in the training set. This way, mean accuracy represents the in-distribution generalization of the model.

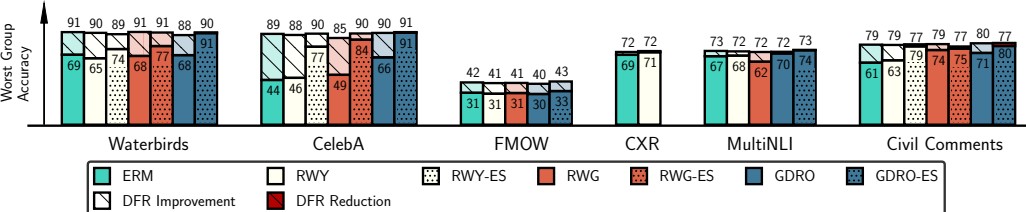

Figure 1: **ERM vs group robustness methods.** Performance of ERM on group robustness methods on vision and NLP benchmark problems. *ES* stands for early stopping. For each method on each dataset we report the base model worst group accuracy (shown with the number inside each bar), and the worst group accuracy after applying DFR (shown above each bar). On CXR-14, we report worst group AUC. While on many datasets the base performance of ERM is much worse compared to group robustness methods, the performance of the different methods is similar after we apply DFR, suggesting that the strong performance of group robustness methods is largely caused by better weighting of the learned features rather than better feature representations.

performance between the methods becomes very small. In fact, on all the datasets and for all the methods the improvement in worst group accuracy from using any of the considered group robustness methods compared to ERM does not exceed 1–2% after applying DFR.

These results suggest that the improvements over ERM in base model performance for methods such as group DRO and RWG are largely the result of better weighting of the learned features rather than learning better representations of the core features. Indeed, if the core feature was better represented by group robustness methods, we would expect to see a significant improvement over ERM after applying DFR.

This observation is significant both practically and scientifically. Practically, the problem of training the last layer is simpler, more data efficient and less compute intensive than retraining the full model [40]. Our results suggest that for many problems, practitioners can primarily focus on retraining the last layer, as training the feature extractor model with group robustness methods does not provide significant improvements. Scientifically, robustness to spurious correlations is often implicitly or explicitly associated with the quality of learned feature representations [e.g. 3, 4, 74, 100, 48, 99]. Our results suggest that the quality of feature representations is not significantly affected by group DRO, refining our understanding of group robustness training and representation learning in the presence of spurious correlations.

**Effect of early stopping.** Early stopping is crucial to achieving strong base model performance with RWY, RWG and group DRO on many of the datasets. In Figure 1 we report the results both with and without early stopping on datasets where the validation worst group accuracy significantly degrades over the course of training. Generally, early stopping does not appear to significantly improve the quality of the learned feature representations even in these problems: after applying DFR, methods with and without early stopping achieve similar performance. This observation suggests that late in training, neural networks may start to assign a higher weight to the spurious features, but the information about the core features is still preserved in the learned representations. For ERM and Group DRO, we explore the DFR WGA performance as a function of the training

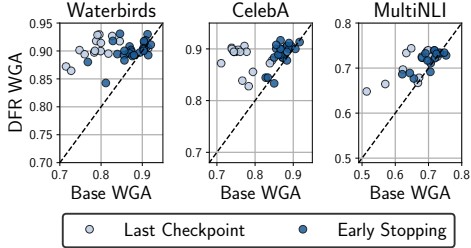

Figure 2: **Group DRO.** Worst group accuracy before and after applying DFR for a range of group DRO runs. DFR does not improve the best runs, indicating that group DRO already learns a nearly optimal last layer.

iteration in Appendix D, also finding that the length of training has a relatively small effect on the final DFR WGA.

**Group DRO analysis.** In Figure 2 we report the worst group accuracy of multiple group DRO runs before and after applying DFR. For each run, we evaluate the best checkpoint according to validation

accuracy (i.e. the checkpoint selected by early stopping) and the last checkpoint saved after a fixed number of epochs. We observe that while the last checkpoints perform poorly in terms of base WGA in most runs, DFR can significantly improve their performance, removing the need for early stopping. Interestingly, we find that the best performing group DRO models cannot be improved by DFR on each of the datasets. This result suggests that the weighting of the features learned by group DRO is already close to optimal, again indicating that the success of group DRO can largely be attributed to learning a better weighting for the features in the last linear layer, rather than learning better features.

> Group robustness methods (e.g. group DRO) perform well because they improve the last linear layer, not the underlying feature representations.

## 6 Effect of the Base Model

Most of the prior work on spurious correlation considers a fixed model class for each problem: for example, on Waterbirds and CelebA datasets almost all the papers use a ResNet-50 base model pre-trained on ImageNet1k [e.g. 76, 50, 48, 40, 84, 63, 100]. Recently, Ghosal et al. [20] showed that vision transformer models [15] may provide better robustness to spurious correlations if pretrained on a large dataset. Here, we perform a systematic large-scale evaluation of the effect of base model choice on the quality of learned feature representations.

We repeat the experiments on the effect of base model, pretraining, and training on the target dataset presented in this section on the MultiNLI text classification task in Appendix E, with similar observations.

### 6.1 Effect of the architecture and pretraining strategy

In Figure 3, we plot the base model mean and worst group accuracy and DFR worst group accuracy for a wide range of models and pretraining strategies on each of the four image classification datasets. We provide model descriptions and training hyper-parameters in Appendix C. We train a total of 78 models on Waterbirds, 78 on CelebA, 40 on FMOW and 40 on CXR.

**Accuracy on the line.** Miller et al. [59] showed that for many distribution shifts in practice the out-of-distribution performance is highly correlated with the in-distribution generalization performance. In Figure 3 (top row), for each dataset we show scatter plots of base model mean accuracy vs base model worst group accuracy, analogously to Miller et al. [59]. For FMOW (which was also considered by Miller et al. [59]) we observe a linear correlation between the base model mean and worst group accuracies. On CXR, the dependence is also roughly linear. However, both on CelebA and on Waterbirds, the correlation does not appear entirely linear. In particular, on Waterbirds there is a large number of models that have similar worst group accuracy $\approx 20\%$, for which there appears to be little correlation between the base model WGA and mean accuracy. On CelebA, the same phenomenon occurs for the best performing models, with WGA between $40\%$ and $50\%$.

**DFR accuracy is on the line.** Next, in the bottom panels of Figure 3, for each of the datasets we report the base model mean accuracy vs DFR WGA. On all datasets other than CXR[3], we observe a high linear correlation between the metrics, including the Waterbirds and CelebA. In particular, the models with the best mean (in-distribution) accuracy also achieve the best DFR WGA. Note that this is not the case for the base model WGA on CelebA, where the best base model WGA is $51\%$ achieved by a ResNet-101 model pretrained on ImageNet1k; this model only achieves mean accuracy of $95.45\%$ compared to $96.2\%$ accuracy for the ConvNext XLarge model. The results in Figure 3 confirm that models that achieve better accuracy on the training data distribution generally learn better representations of the core features, and provide better worst group accuracies with DFR.

**Are VITs more robust than CNNs?** Ghosal et al. [20] noted that vision transformers pre-trained on ImageNet22k achieved better worst group accuracies on benchmarks with spurious correlations than popular CNN models. In particular, with a VIT-B/16 model, they achieve $89.3\%$ worst group accuracy on Waterbirds. With the ConvNext Large model pretrained on ImageNet22k with ImageNet1k

---

[3]On CXR, we found that the base model is often hard to improve with DFR, suggesting that it already learns to weight the features correctly. We discuss the possible reasons and the nature of spurious features on this dataset in Appendix A.2.

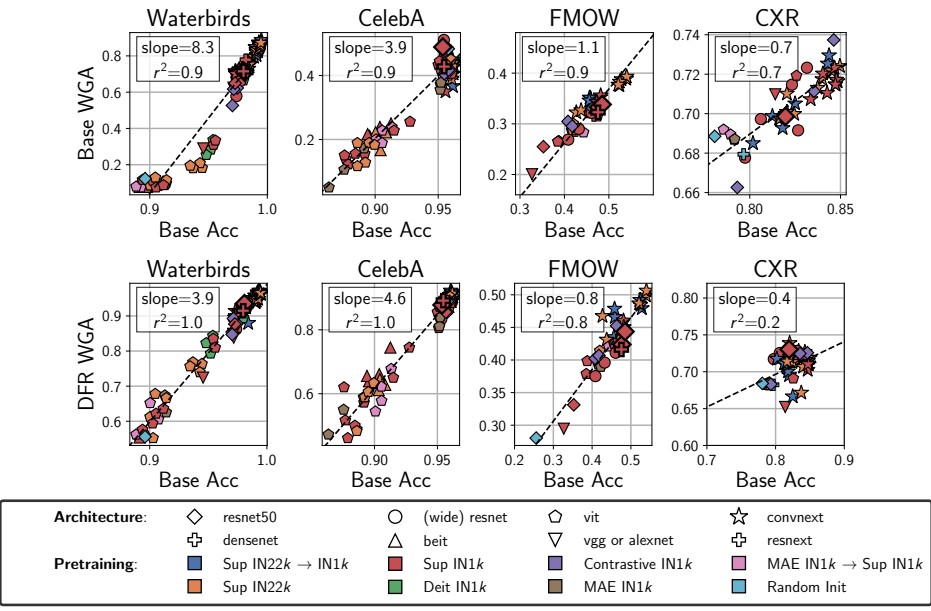

Figure 3: **Effect of the model architecture and pretraining.** Base model in-distribution accuracy plotted against base worst-group accuracy (top row) and DFR worst group accuracy (bottom row). For each panel, we additionally estimate the slope and the $r^2$ score for the linear fit to the data. On all datasets other than CXR, the DFR WGA is linearly correlated with base model in-distribution performance. For the base model WGA, the correlation with in-distribution performance is not entirely linear. Generally, models with better in-distribution performance provide better worst-group performance and better core feature representations.

finetuning, we achieve $88.9\%$ worst group accuracy on the Waterbirds dataset. Notably, ConvNext is a CNN model and *not a vision transformer*. Generally, we observe that the models that provide the best in-distribution performance also provide better WGA. In our experiments, we did not observe qualitative differences between the results for vision transformers and CNN models.

**ERM features are sufficient for SOTA performance.** The DFR WGA results for the Waterbirds, CelebA and FMOW datasets significantly improve upon the previous best reported results, to the best of our knowledge. In particular, the ConvNext Large model pretrained on ImageNet22k with ImageNet1k finetuning achieves $97.2\%$ DFR WGA on Waterbirds and $92.2\%$ on CelebA; on FMOW, we only considered smaller ConvNext versions due to computational constraints, still achieving $50.6\%$ DFR WGA with ConvNext Small pretrained on ImageNet22k; the current best results on the WILDS leaderboard for this dataset are $47.6\%$ followed by $35.5\%$ [43]. We note that our DFR evaluation uses the validation set to train the last layer of the model, similarly to e.g. Nam et al. [63], and unlike most standard group robustness methods which only use the validation set to tune the parameters. However, the results presented in this section prove that standard ERM with a strong pretrained model can achieve outstanding results on the robustness benchmarks, significantly improving upon specialized group robustness methods using a weaker model.

> Strong in-distribution generalization correlates with improved robustness to spurious correlations (measured by the DFR worst group accuracy), and this trend holds regardless of the underlying architecture (CNN or ViT).

### 6.2 Does training on the target data improve features?

Above, we have shown that the choice of the base model architecture and pretraining has a large effect on the quality of the learned feature representations, as measured by DFR WGA. It is then natural to ask how much of the feature learning happens during training on the target data, and how much is simply transferred from the pretraining task. To answer this question, for the models trained

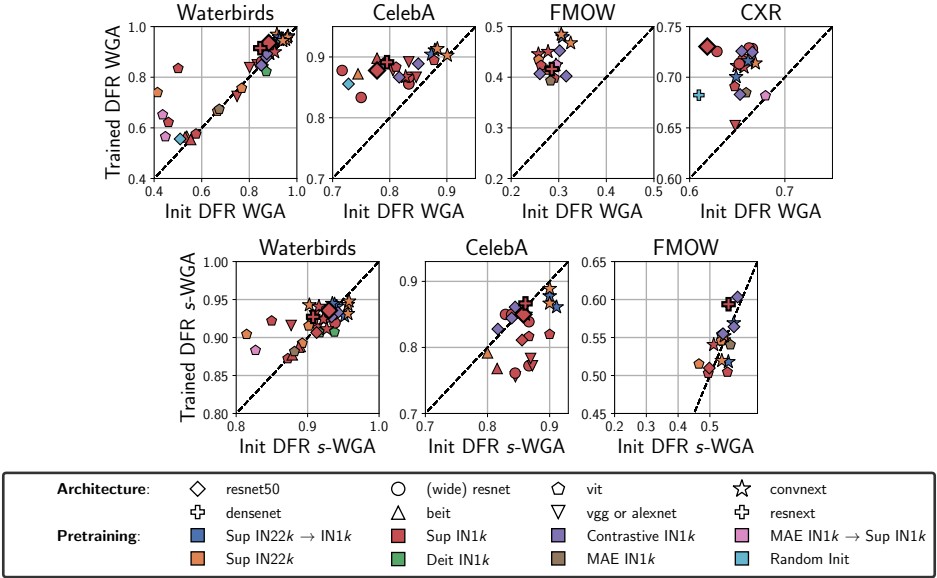

Figure 4: **Effect of training on target data.** DFR WGA (top 4 panels) and DFR $s$-WGA (bottom 3 panels) before and after training. On Waterbirds, the results do not significantly improve from training on the target data. On the other datasets, the performance improves consistently after training on the target data. Interestingly, on CelebA the DFR $s$-WGA decreases during training for many models, meaning that the spurious gender feature becomes less predictable from the learned representations.

in the previous section, we run the DFR evaluation on the initial weights of the models, without training the feature extractor on the target data. We report the DFR WGA and DFR $s$-WGA results in Figure 4. We repeat this experiment on the MultiNLI dataset in Appendix Table 5.

Surprisingly, we find that on Waterbirds, for most models the improvement from training on the target (Waterbirds) data is very small, if any. For example, for the ImageNet1k-pretrained ResNet-50 model that was not trained on Waterbirds data at all[4], we get $88.2\%$ worst group accuracy by simply training the last layer on the validation data with DFR. If we finetune the feature extractor on the Waterbirds training data, we can achieve $92.9\%$ DFR WGA. For reference, the state-of-the-art group DRO method achieves $91\%$ WGA on Waterbirds with this architecture.

Furthermore, with the ConvNext Large model pretrained on ImageNet22k, we get $94\%$ DFR WGA without training the feature extractor on the Waterbirds data, exceeding the best results previously reported in the literature, to the best of our knowledge. This strong performance is not particularly surprising, as ImageNet22k has many of the Waterbirds bird types as classes. The performance is almost unchanged by training on the target data: DFR WGA after training is $94.3\%$. From these results, we can conclude that Waterbirds performance should not be used as a primary metric for feature learning performance, especially if large-scale pretraining is used! Indeed, it is possible to achieve outstanding performance, exceeding the previously reported state-of-the-art, without training the features on the target data.

On the other datasets, the feature learning is more pronounced: the DFR WGA improves after training for all the models considered. However, on CelebA it is still possible to achieve $88.3\%$ WGA without training on CelebA data, with ConvNext XLarge pretrained on ImageNet22k, while the best result that we were able to achieve with feature extractors trained on CelebA is $92.2\%$.

**Is the spurious feature representation improved during training?** We additionally explore the quality of representation of the spurious feature via DFR $s$-WGA. We show the results in Figure 4 (bottom row). Interestingly, on CelebA the spurious gender feature becomes less predictable during training for many of the models. On FMOW and Waterbirds there is no consistent trend, and the spurious feature does not become significantly more or less predictable during training.

---

[4]This model is used as initialization by most works that report experiments on the Waterbirds dataset.

> Strong feature extractors trained on large-scale datasets are sufficient for outstanding performance on benchmark spurious correlation datasets, without any finetuning on the target data. Finetuning can improve the quality of the learned representations but often only to a relatively small degree (e.g. Waterbirds).

## 6.3 Effect of pretraining strategy

Finally, using a fixed ResNet-50 model architecture we evaluate the effect of pretraining strategy. On each of the four datasets, we train a ResNet-50 model initialized with (1) random initialization, (2) supervised pretraining, (3) DINO pretraining [10], (4) SimCLR pretraining [11] and (5) Barlow Twins pretraining [96] on ImageNet1k. We report the results in Figure 5. On Waterbirds and FMOW datasets, the randomly initialized model does not provide competitive performance. On CelebA, it still underperforms the pretrained models, but the gap is much smaller. Among all pretraining methods, supervised is preferable, but the contrastive methods are highly competitive.

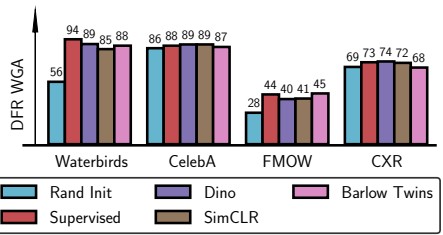

Figure 5: **Effect of pretraining.** Supervised pretraining provides the best performance, but contrastive methods are competitive. Random initialization performs poorly.

In Appendix Table 4 on MultiNLI with BERT models, we also show that pretraining is crucial for strong performance, while the specific choice of pretraining data has a smaller effect. In Appendix D we provide further ablations exploring the effect of weight decay and data augmentation on the learned feature representations. We find that while appropriate regularization can be helpful, models trained with no weight decay and no data augmentation can achieve highly competitive DFR WGA, especially on the standard Waterbirds and CelebA benchmarks.

> ImageNet pretraining improves the representation of core features on many image datasets with spurious correlations.

## 7 Discussion

The worst group performance of a model is affected by two factors: the quality of the representation of the core features produced by the feature extractor and the weight assigned to the core features in the last classification layer. In contrast to prior work, we consider the quality of the feature extractor in isolation, focusing on realistic datasets and large-scale models. We find that many of the popular group robustness methods improve the worst group performance primarily by learning a better last layer and not by learning a better feature representation. Similarly, regularization techniques such as early stopping and strong weight decay can improve the worst group accuracy by learning a better last layer, but do not lead to a consistent improvement in terms of the quality of the learned feature representations. On the other hand, the base model architecture and pre-training strategy have a major effect on the quality of the feature representations.

Our observations suggest an important open question: is it possible to significantly improve upon standard ERM in terms of the quality of the learned representations for a given base model? In the future work, we will evaluate methods such as Rich Feature Construction [99], gradient starvation [68], ensembling [47] and other techniques for increasing feature diversity [e.g. 48]. We hope that our work will also inspire new group robustness methods targeted specifically at improving the quality of the core feature representations.

**Acknowledgements.**    This research is supported by NSF CAREER IIS-2145492, NSF I-DISRE 193471, NIH R01DA048764-01A1, NSF IIS-1910266, NSF 1922658 NRT-HDR: FUTURE Foundations, Translation, and Responsibility for Data Science, NSF Award 1922658, Meta Core Data Science, Google AI Research, BigHat Biosciences, Capital One, and an Amazon Research Award. This work was supported in part through the NYU IT High Performance Computing resources, services, and staff expertise.

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
