## Appendix Outline

This appendix is structured as follows. In Section A we describe the datasets, augmentation policies and models used in this paper. In Section B we provide details on the methods, implementations and hyper-parameters, as well as detailed results for the experiments in Section 5. In Section C we provide additional details on the experiments in Section 6. In Section D we provide results on the effect of regularization and data augmentation. In Section E we provide additional results on the MultiNLI dataset. Finally, in Section F we describe the limitation, broader impact, compute and licenses.

**Tools and packages.** During the work on this paper, we used the following tools and packages: `NumPy` [24], `SciPy` [87], `PyTorch` [67], `TorchVision` [55], `Jupyter notebooks` [42], `Matplotlib` [33], `Pandas` [57], `Weights&Biases` [7], `timm` [90], `transformers` [92], `vissl` [21].

## A  Data and Models

In this section, we describe the datasets, data augmentation policies and models used throughout the paper.

### A.1  Datasets

We perform experiments on 4 image classification and 2 text classification problems. We illustrate the image datasets in Figure 6 and the text datasets in Figure 7.

**Waterbirds.**  The Waterbirds dataset is described in Figure 6. The dataset contains images of birds from the CUB dataset [88] pasted on the backgrounds from the Places dataset [102]. The spurious attribute $s$ describes the type of background (water or land) and the core feature associated with the target $y$ is the type of the bird (waterbird or landbird). For a detailed description of the data generating process, see Sagawa et al. [76]. The background is spuriously correlated with the bird type in the training datasets: waterbirds are more likely to be placed on a water background, and landbirds are more likely to be placed on a land background. There are 4 groups defined to the tuples $(y, s)$.

**CelebA.**  The CelebA hair color dataset is described in Figure 6. The dataset contains photos of celebrities from the CelebA dataset Liu et al. [51]. The core attribute associated with the target $y$ is the hair color (blond vs non-blond). The gender serves as a spurious feature $s$: the vast majority of blond people in CelebA are female. There are 4 groups defined to the tuples $(y, s)$.

**FMOW.**  The WILDS-FMOW dataset is described in Figure 6. This dataset is a part of the WILDS benchmark [43, 77], and was originally collected by Christie et al. [12]. The dataset contains satellite images, and the target $y$ describes the type of building or land use shown in the image. There are 62 classes. The spurious attribute $s$ corresponds to the region (Asia, Europe, Africa, America, Oceania) shown in the image. The training data additionally contains another group *Other*, which is dropped during evaluation. For this dataset, following the WILDS benchmark, we define groups by just the value of the spurious attribute: $g = s$. In particular, worst group accuracy corresponds to the worst accuracy across regions. The regions are represented unequally in the data, leading to unequal performance. Moreover, the test images were taken several years later than the train images, constituting an additional type of distribution shift. We note that for the DFR evaluation, we use the validation data to train the last layer of the model, which means that our results would be classified as non-standard submissions to the WILDS leaderboard.

**CXR.**  The CXR dataset is described in Figure 6. The images are taken from the CXR-14 dataset [89]. CXR-14 is a multi-label dataset, where each label corresponds to a disease, and one image can show multiple diseases. We perform the pneumothorax classification task, i.e. all images that have the pneumothorax label have $y = 1$ and all images that do not have $y = 0$. The dataset contains several images for some of the patients; there is no patient overlap between the train, test and validation splits. Oakden-Rayner et al. [65] identified a hidden stratification in this dataset: a lot of images showing patients with pneumothorax showed a *chest drain*, which is a treatment for the pneumothorax disease. The neural networks trained on this dataset are using the chest drain as a shortcut feature, and perform much worse when classifying images without the chest drain. The labels for the spurious feature are only available on the validation and test datasets, and only for the images showing sick patients, i.e. there is no $(y = 0, s = 1)$ group. There are 3 groups corresponding to available pairs $(y, s)$.

Following prior work [e.g. 65, 48, 70], we compute three AUC values: for classifying group 0 against group 1, group 0 against group 2 and group 0 against the combined groups 0 and 1; instead of worst group accuracy we report the lower of the first two AUC values, and instead of mean accuracy we report the last AUC value throughout the experiments.

**Civil Comments**  The Civil Comments Coarse dataset is described in Figure 7. The dataset was originally collected in Borkan et al. [8] and is a part of the WILDS benchmark [43, 77]. This dataset contains comments that are classified as toxic or not toxic. We use the coarse version of the dataset, following Idrissi et al. [34]. The spurious attribute $s$ determines whether or not the comment mentions one of the following protected attributes: male, female, LGBT, black, white, Christian, Muslim, other religions. These protected attributes are mentioned more frequently in toxic comments compared to neutral comments, constituting a spurious correlation. There are 4 groups corresponding to the pairs $(y, s)$.

**MultiNLI**  The MultiNLI dataset is described in Figure 7. It contains pairs of sentences, and the label $y$ describes the relationship between the sentences: contradiction, entailment or neutral. The spurious attribute $s$ describes the presence of negation words, which appear much more frequently in the examples from the negation class. There are 6 groups corresponding to the pairs $(y, s)$.

## A.2 The nature of the spurious correlations

The nature of the spurious correlation differs between the datasets that we consider. On Waterbirds, CelebA, Civil Comments and MultiNLI, the spurious attribute is correlated with the target on train, but is not generally predictive of the class label across the groups. In particular, we wish to train a model that ignores the spurious attribute in its predictions, as using the spurious attribute *hurts* the performance on some of the groups. By applying DFR on a group-balanced validation set, we try to train a model that ignores the spurious feature.

On the FMOW dataset, the groups correspond to the spurious attribute (region), and not pairs $(y, s)$. As the regions are not represented equally, standard ERM is incentivised to perform well on the majority groups, with less weight on the minority groups. In this case, we do not wish to remove the reliance on the spurious attribute, but instead we wish to find a model that performs well on the minority groups. We apply DFR to this dataset by retraining the last layer on a group-balanced validation set, analogously to the other datasets.

Finally, in the CXR dataset, there are no examples showing healthy patients with a chest drain (to the best of our knowledge). Consequently, the reliance on the spurious attribute $s$ does not necessarily *hurt* performance on the other groups, unlike e.g. on the Waterbirds dataset. In this case, we do not wish to remove ignore the feature $s$, but instead we wish to find a model that performs well on the images with $s = 0$ (no chest drain). Applying DFR to this dataset is not straightforward, as (1) we do not have examples from the $(y = 0, s = 1)$ group, and (2) we do not wish to remove the reliance on the $s$ attribute. We found that the best approach in this case was to simply train a logistic regression model on all of the available validation data, without any balancing. DFR generally provides a smaller improvement on this dataset, and behaves less consistently, as we see e.g. in Figure 3.

| **Waterbirds** | | **Target**: bird type; | | **Spurious feature**: background type. |
|---|---|---|---|---|
| **Image:** |  |  |  |  |
| **Group $g$:** | 0 | 1 | 2 | 3 |
| **Target $y$:** | 0 | 0 | 1 | 1 |
| **Spurious $s$:** | 0 | 1 | 0 | 1 |
| **Description:** | landbird on land | landbird on water | waterbird on land | waterbird on water |
| **# Train data:** | 3498 (73%) | 184 (4%) | 56 (1%) | 1057 (22%) |
| **# Val data:** | 467 | 466 | 133 | 133 |

| **CelebA hair color** | | **Target**: hair color; | | **Spurious feature**: gender. |
|---|---|---|---|---|
| **Image:** |  |  |  |  |
| **Group $g$:** | 0 | 1 | 2 | 3 |
| **Target $y$:** | 0 | 0 | 1 | 1 |
| **Spurious $s$:** | 0 | 1 | 0 | 1 |
| **Description:** | non-blond woman | non-blond man | blond woman | blond man |
| **# Train data:** | 71629 (44%) | 66874 (41%) | 22880 (14%) | 1387 (1%) |
| **# Val data:** | 8535 | 8276 | 2874 | 182 |

| **Wilds-FMOW** | | **Target**: land use / building; | | **Spurious feature**: region. | |
|---|---|---|---|---|---|
| **Image:** |  |  |  |  |  |
| **Group $g$:** | 0 | 1 | 2 | 3 | 4 |
| **Target $y$:** | $\{0,\dots,61\}$ | $\{0,\dots,61\}$ | $\{0,\dots,61\}$ | $\{0,\dots,61\}$ | $\{0,\dots,61\}$ |
| **Spurious $s$:** | 0 | 1 | 2 | 3 | 4 |
| **Description:** | Asia | Europe | Africa | America | Oceania |
| **# Train data:** | 17809 (23%) | 34816 (45%) | 1582 (2%) | 20973 (27%) | 1641 (2%) |
| **# Val data:** | 4121 | 7732 | 803 | 6562 | 693 |

| **CXR-14** | **Target**: pneumothorax; | | **Shortcut feature**: chest drain. |
|---|---|---|---|
| **Image:** |  |  |  |
| **Group $g$:** | 0 | 1 | 2 |
| **Target $y$:** | 0 | 1 | 1 |
| **Spurious $s$:** | 0 | 0 | 1 |
| **Description:** | not sick, no chest drain | sick, no chest drain | sick, chest drain |
| **# Train data:** | 71629 (95%) | ? | ? |
| **# Val data:** | 10714 | 204 | 300 |

Figure 6: **Image datasets.** Group descriptions and example images for Waterbirds, CelebA hair color, Wilds-FMOW and CXR-14 datasets. Each column corresponds to a group in the dataset. For CXR-14 group labels are not known on the training data.

**Civil Comments**  **Target**: toxic / neutral comment;
**Spurious feature**: mentions protected categories.

| Example | Group $g$ | Target $y$ | Spur. $s$ | # Train data | # Val data |
|---|---|---|---|---|---|
| *"I wouldn't think this would be so rare on the plains of eastern Colorado."* | 0 | 0 (Netral) | 0 | 148186 (55%) | 25159 |
| *"If the person wanted to write to the Bishop, he or she would have. They wanted the Vatican to know their pain, not to acknowledge it is a lack of Christian charity and kindness."* | 1 | 0 (Netral) | 1 | 90337 (33%) | 14966 |
| *"What a gross example of bureaucrats and lawyers showing everyone in Oregon who our bosses are. Next the jerks will demand we bow to them. <...>"* | 2 | 1 (Toxic) | 0 | 12731 (5%) | 2111 |
| *"Democrats, RINO's and atheists won't be happy until they have destroyed conservatives and christians."* | 3 | 1 (Toxic) | 1 | 17784 (7%) | 2944 |

**MultiNLI**  **Target**: contradiction / entailment / neutral;
**Spurious feature**: has negation words.

| Example | Group $g$ | Target $y$ | Spur. $s$ | # Train data | # Val data |
|---|---|---|---|---|---|
| *"he was up quickly. [SEP] he sat the entire time and didn't move."* | 0 | 0 (contr.) | 0 | 57498 (28%) | 22814 |
| *"for golf enthusiasts, two courses reassure the visitor you are close to civilization. [SEP] there is no golf course available anywhere around."* | 1 | 0 (contr) | 1 | 11158 (5%) | 4634 |
| *"while emergency physicians may not have the time or interest, the patients do. [SEP] the patients have time and interest, unlike emergency physicians."* | 2 | 1 (entail.) | 0 | 67376 (32%) | 26949 |
| *"then it dawned on him that of course the lawyer did not know. [SEP] he realised that the lawyer had no idea."* | 3 | 1 (entail.) | 1 | 1521 (1%) | 613 |
| *"disneyland is huge and can be very crowded in summer. [SEP] going to disneyland is every child's dream.* | 4 | 2 (neutr.) | 0 | 66630 (32%) | 26655 |
| *"you have raced him, senor? " he asked drew with formal courtesy. [SEP] drew replied that he had never raced him."* | 5 | 2 (neutr.) | 1 | 1992 (1%) | 797 |

Figure 7: **Text datasets.** Text examples, class labels, spurious attributes and group labels for the Civil Comments and MultiNLI datasets. On both datasets, the spurious feature is correlated with the class label.

### A.3 Data augmentation and preprocessing

On the image datasets, we consider several data augmentation policies. In Figure 8, we provide the code implementing the **No augmentation** and **Default augmentation** policies in torchvision. The other policies are more involved, so we do not include the code here. All policies normalize the data analogously to the code in Figure 8.

**No augmentation.** This policy resizes the data to a fixed resolution and applies channel normalization.

**Default policy.** This policy additionally applies random crops and horizontal flips.

**MixUp.** For this policy, we use the Default policy to initially preprocess the images, and then apply MixUp [98] with the mixing parameter $\alpha = 0.2$.

**Random Erasing.** This policy is described in Zhong et al. [101]. It randomly erases rectangular blocks of the image, and replaces them with uniform grey blocks. We use the implementation in `timm.data.random_erasing` in the `timm` package.

**Augmix.** We adapt the official implementation of the AugMix policy [29] available here.

**Text models.** For data preprocessing in the experiments on text data, we use the BERT tokenizer: `BertTokenizer.from_pretrained("bert-base-uncased")` from the `transformers` package.

### A.4 Models

Here, we list the models used in this paper.

**ResNet-50.** On the Waterbirds, CelebA and FMOW by default we use the ResNet-50 model He et al. [26] pretrained on ImageNet. We use the model implemented in the `torchvision` package: `torchvision.models.resnet50(pretrained=True)`. In Figure 5, we additionally consider the randomly initialized model `torchvision.models.resnet50(pretrained=False)`, and the models pre-trained with SimCLR [11] and Barlow Twins [96] contrastive learning methods, imported from the `vissl` package (models available here).

**DenseNet-121.** On CXR, by default we use the DenseNet-121 model [32] implemented in the `torchvision` package: `torchvision.models.densenet121(pretrained=True)`. We also consider this model without ImageNet pretraining: `torchvision.models.densenet121(pretrained=False)`.

**Other image models.** In Section 6, we additionally consider a broad range of architectures and pretraining methods, which we briefly list here. We use the following models from `torchvision`, all pretrained on ImageNet1k: ResNet-18, ResNet-34, ResNet-50, ResNet-101, ResNet-152 [26]; Wide-ResNet-50-2 [95]; ResNext-50-32 $\times$ $4d$ [94]; DenseNet-121 [32]; VGG-16, VGG-19 [82]; AlexNet [46]. We also use the following models from the `timm` package: ConvNext-Small, ConvNext-Base, ConvNext-Large, ConvNext-XLarge [52], pretrained on either ImageNet1k or ImageNet22k; ViT-Small, ViT-Base, ViT-Large, ViT-Huge [15], pretrained on either ImageNet1k or ImageNet22k; BEiT-Base, BEiT-Large [6], pretrained on either ImageNet1k or ImageNet22k; DEiT-Small, DEiT-Base [6], pretrained ImageNet1k. We also use ViT-Small, ViT-Base models with DINO pretraining on ImageNet1k [10] available here. Finally, we use ViT-Base, ViT-Large and ViT-Huge models with MAE pretraining on ImageNet1k [25], with or without supervised finetuning on ImageNet1k, available here.

**BERT model.** On the text classification problems, we use the BERT for classification model from the `transformers` package: `BertForSequenceClassification.from_pretrained('bert-base -uncased', num_labels=num_classes)`.

## B  Details: ERM vs Group Robustness

### B.1 Methods and hyper-parameters

**ERM, RWY and RWG.** We report the hyper-parameters used on each of the datasets in Table 2. We did not tune the hyper-parameters for ERM, RWY and RWG aside from the learning rate for the text classification problems. For all vision datasets, we used the default data augmentation policy (see Section A.3). We describe the default model choices in Section A.4. The RWY method is

```python
import torchvision.transforms as transforms
import torch

target_resolution = (224, 224)
resize_resolution = (256, 256)
IMAGENET_STATS = ([0.485, 0.456, 0.406], [0.229, 0.224, 0.225])

# No Augmentation
noaug_transform = transforms.Compose(
    transforms.RandomResizedCrop(
        target_resolution,
        scale=(0.7, 1.0),
        ratio=(0.75, 1.33),
        interpolation=2),
    transforms.RandomHorizontalFlip(),
    transforms.ToTensor(),
    transforms.Normalize(*IMAGENET_STATS)
    )

# Default Augmentation
aug_transform = transforms.Compose(
    transforms.Resize(resize_resolution),
    transforms.CenterCrop(target_resolution)
    transforms.ToTensor(),
    transforms.Normalize(*IMAGENET_STATS)
    )

# On test, we always use noaug_transform
test_transform = noaug_transform
```

Figure 8: **Data augmentation policies.** The default data augmentation policies implemented using the `torchvision` package.

implemented by providing a `sampler` to the `DataLoader` in `PyTorch`, which samples the datapoints from different classes with the same frequency. The RWG method is implemented analogously, and samples datapoints from different groups with the same frequency.

**Group DRO.** The training objective used in Sagawa et al. [76] is the following:

$$\hat{\theta} = \arg \min_{\theta} \max_{g \in \mathcal{G}} \left[ \mathbb{E}_{(x,y) \sim p_g} l(y, f_\theta(x)) + C/\sqrt{n_g} \right],$$

where $f_\theta(\cdot)$ is a neural network model with parameters $\theta$, $l(\cdot, \cdot)$ is a loss function (cross-entropy for classification), $\mathcal{G}$ is a set of all groups, $n_g$ is the size of the group $g$, and $C$ is a generalization adjustment hyper-parameter. We follow Sagawa et al. [76] to choose hyper-parameter combinations for tuning group DRO on Waterbirds, CelebA and MultiNLI.

In particular, on Waterbirds we considered the following combinations of the initial learning rate $lr$ and weight decay $wd$: $(lr = 10^{-3}, wd = 10^{-4}), (lr = 10^{-4}, wd = 0.1)$ and $(lr = 10^{-5}, wd = 1)$. We varied the parameter $C$ in the range $\{0, 1, 2, 3, 4, 5\}$ for each combination of learning rate and weight decay. Additionally, we varied weight decay in range $\{0, 10^{-4}, 10^{-4}, 3 \cdot 10^{-4}, 10^{-3}, 3 \cdot 10^{-3}, 10^{-2}, 3 \cdot 10^{-2}, 0.1, 0.3, 1\}$ for fixed values of learning rate $10^{-3}$ and $C = 0$.

On CelebA, we considered the following combinations of $lr$ and $wd$: $(lr = 10^{-4}, wd = 10^{-4}), (lr = 10^{-4}, wd = 10^{-2})$ and $(lr = 10^{-5}, wd = 0.1)$, varying $C$ in the same range as on Waterbirds. For weight decay ablation we considered values in $\{0, 10^{-5}, 3 \cdot 10^{-5}, 10^{-4}, 3 \cdot 10^{-4}, 10^{-3}, 3 \cdot 10^{-3}, 10^{-2}\}$.

On MultiNLI, we use learning rate $2 \cdot 10^{-5}$, and vary weight decay in the range $\{0., 0.01, 0.1, 1.0\}$ and $C$ in the range $\{0, 1, 3, 5\}$.

| Method | Waterbirds | CelebA | FMOW | CXR | MultiNLI | CivilComments |
|---|---|---|---|---|---|---|
| **ERM** | $68.9_{\pm 2.0}$ | $44.0_{\pm 2.1}$ | $31.4_{\pm 0.7}$ | $68.8_{\pm 0.2}$ | $67.5_{\pm 1.1}$ | $61.0_{\pm 0.3}$ |
| **ERM + DFR** | $91.1_{\pm 0.8}$ | $89.4_{\pm 0.9}$ | $41.6_{\pm 0.6}$ | $71.6_{\pm 0.5}$ | $72.6_{\pm 0.3}$ | $78.8_{\pm 0.5}$ |
| **RWY** | $65.4_{\pm 0.6}$ | $46.1_{\pm 2.1}$ | $30.5_{\pm 0.6}$ | $71.3_{\pm 1.1}$ | $68.0_{\pm 0.4}$ | $63.4_{\pm 0.9}$ |
| **RWY + DFR** | $90.4_{\pm 1.0}$ | $88.3_{\pm 0.5}$ | $40.9_{\pm 0.7}$ | $72.5_{\pm 0.1}$ | $72.2_{\pm 1.9}$ | $78.5_{\pm 0.4}$ |
| **RWY-ES** | $74.5_{\pm 0.0}$ | $76.8_{\pm 7.7}$ | | | | $78.9_{\pm 1.0}$ |
| **RWY-ES + DFR** | $89.1_{\pm 0.7}$ | $89.6_{\pm 0.5}$ | | | | $76.9_{\pm 0.6}$ |
| **RWG** | $67.7_{\pm 0.7}$ | $49.1_{\pm 0.9}$ | $31.2_{\pm 0.1}$ | | $62.0_{\pm 0.2}$ | $73.5_{\pm 2.2}$ |
| **RWG + DFR** | $91.3_{\pm 0.3}$ | $85.4_{\pm 1.5}$ | $41.1_{\pm 0.6}$ | | $71.6_{\pm 1.3}$ | $79.3_{\pm 0.5}$ |
| **RWG-ES** | $77.4_{\pm 0.0}$ | $83.7_{\pm 0.7}$ | | | | $74.7_{\pm 6.7}$ |
| **RWG-ES + DFR** | $90.7_{\pm 0.2}$ | $89.8_{\pm 0.3}$ | | | | $77.0_{\pm 0.2}$ |
| **GDRO** | $68.5_{\pm 6.0}$ | $66.3_{\pm 7.8}$ | $30.2$ | | $70.1$ | $70.6$ |
| **GDRO + DFR** | $88.2_{\pm 1.1}$ | $90.4_{\pm 0.7}$ | $40.3$ | | $71.8$ | $80.2$ |
| **GDRO-ES** | $90.7_{\pm 0.6}$ | $90.6_{\pm 1.6}$ | $33.1$ | | $73.5$ | $80.4$ |
| **GDRO-ES + DFR** | $89.9_{\pm 0.5}$ | $91.1_{\pm 0.1}$ | $42.5$ | | $73.3$ | $77.3$ |

Table 1: **Method comparison results.** Detailed results for the method comparison presented in Figure 1. The error bars represent one standard deviation over 3 independent runs. We only evaluate early stopping (ES) on datasets where it is helpful for the base model performance. On CXR, no group information is available on the train data, so we can only apply the ERM and RWY methods. We use a ResNet50 model pretrained on ImageNet on Waterbirds, CelebA and FMOW, a DenseNet-121 model pretrained on ImageNet on CXR, and a BERT model pretrained on Book Corpus and English Wikipedia data on CivilComments and MultiNLI.

| Dataset | Optimizer | Initial LR | LR schedule | Batch size | Weight decay | # Epochs |
|---|---|---|---|---|---|---|
| **Waterbirds** | SGD [71] | $3 \cdot 10^{-3}$ | Cosine annealing | 32 | $10^{-4}$ | 100 |
| **CelebA** | SGD [71] | $3 \cdot 10^{-3}$ | Cosine annealing | 100 | $10^{-4}$ | 20 |
| **FMOW** | SGD [71] | $3 \cdot 10^{-3}$ | Cosine annealing | 100 | $10^{-4}$ | 20 |
| **CXR** | SGD [71] | $3 \cdot 10^{-3}$ | Cosine annealing | 100 | $10^{-4}$ | 20 |
| **MultiNLI** | AdamW [53] | $10^{-5}$ | Linear annealing | 16 | $10^{-4}$ | 10 |
| **Civil Comments** | AdamW [53] | $10^{-5}$ | Linear annealing | 16 | $10^{-4}$ | 10 |

Table 2: **ERM, RWY and RWG hyper-parameters.** Default hyper-parameters used on each dataset. On the image classification datasets, we adapted the hyper-parameters of Kirichenko et al. [40], with no tuning. On the text classification dataset, we followed Sagawa et al. [76] and Idrissi et al. [34] in the choice of the optimizer and learning rate scheduler, and chose the learning rate value which provided the best base model validation mean accuracy.

On CivilComments, we considered the following combinations of $lr$ and $wd$: $(lr = 10^{-5}, wd = 0.1), (lr = 10^{-5}, wd = 0.01), (lr = 10^{-4}, wd = 0.01)$; for each combination we varied $C$ in $\{0, 3, 5\}$.

On FMOW, the learning rate and weight decay pairs were $(lr = 10^{-3}, wd = 10^{-3})$, $(lr = 10^{-4}, wd = 10^{-3})$, $(lr = 10^{-4}, wd = 10^{-2})$, $(lr = 10^{-5}, wd = 10^{-1})$. For each $lr$ and $wd$ combination we varied $C$ in the same range as on Waterbirds and CelebA datasets.

For all image datasets, we use the default data augmentation policy (see Section A.3). In all runs we used the same batch size and train for the same number of epochs as the corresponding hyper-parameters in ERM (see Table 2).

## B.2 DFR implementation and hyper-parameters

In all experiments, we use the $\text{DFR}_{\text{Tr}}^{\text{Val}}$ variation of the DFR method described in Kirichenko et al. [40]. We follow the official implementation provided here. Specifically, we use $\ell_1$ regu-

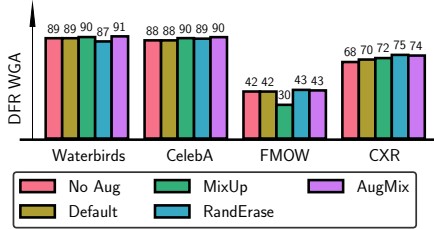

Figure 9: **Effect of data augmentation.** All the results shown use a ResNet50 architecture except for CXR results, which use a DenseNet-121 model pretrained on ImageNet. AugMix augmentation policy consistently provides strong performance across all datasets, while random erasing and mixup hurt performance on Waterbirds and FMOW respectively.

larization for training the logistic regression model implemented in the `scikit-learn` package: `sklearn.LogisticRegression(penalty="l1", C=c, solver="liblinear")`; we tune the regularization strength $c$ within the set $\{1, 0.7, 0.3, 0.1, 0.07, 0.03, 0.01\}$, following the procedure described in Kirichenko et al. [40].

As explained in Section A.2, on CXR we train the logistic regression model on all of the validation set without group balancing. Further, on CXR we tune the regularization strength parameter according to the worst AUC, and not worst group accuracy.

We additionally compute DFR $s$-WGA by using DFR (with the same hyper-parameters and implementation) to predict the spurious attribute $s$ (instead of the class label $y$) from the learned features.

### B.3 Results

We provide detailed results for all methods on all datasets in Table 1. Group-DRO significantly improves the base model performance compared to all other methods across the board. After applying DFR, the performance across the different methods is very similar, although Group-DRO with early stopping still typically provides a small improvement. The improvement is, however, very small compared to the improvement from using a better base model (see Section 6).

## C  Details: Effect of the Base Model

For all experiments in Section 6, we use the default data augmentation policy (see Section A.3). We consider a broad range of models and architectures (see Section A.4). We ran all models with the default hyper-parameters provided in Table 2. For CXR-14 dataset, we used class reweighting due to heavy class imbalance present in the train data (95% of train images are from the negative class).

We note that the default hyper-parameters are suboptimal for some of the ViT models, which lead to poor performance for some of the models. With more tuning, we expect that it should be possible to improve the results further for all the considered models, especially ViT-based.

## D  Effect of Regularization

Regularization is used to combat overfitting, including reliance on spurious features. We consider two regularization techniques: weight decay and data augmentation.

**Effect of weight decay.**  Using a ResNet-50 model pretrained on ImageNet1k, we run training with a range of weight decay values on each of the four image datasets, and report the results in Figure 10. We use the default model setup described in Section A.4, and default hyper-parameters in Table 2, and vary the weight decay strength in the range $\{0, 10^{-5}, 10^{-4}, 3 \cdot 10^{-4}, 10^{-3}, 3 \cdot 10^{-3}, 10^{-2}\}$. For the base model WGA performance, it is generally helpful to set the weight decay to non-zero values (in particular, on Waterbirds, CelebA and FMOW datasets). Indeed, on CelebA the base model WGA for no weight decay is the worst across all weight decay values, losing to the best weight decay by $\approx 5\%$. However, the no weight decay model is in fact the best model according to DFR WGA

on CelebA! On Waterbirds and CelebA, strong weight decay allows the model to rely less on the spurious features in the last layer (leading to higher base WGA), but does not improve the learned feature representations (similar or worse DFR WGA). This observation is especially interesting given that Sagawa et al. [76] showed that group DRO requires stronger than usual weight decay to achieve good performance.

At the same time, on FMOW and CXR non-zero weight decay appears to be helpful for learning high quality representations of the core features. We hypothesize that the difference in weight decay effects on these two pairs of datasets is due to the difference in the nature of distribution shifts. In particular, Waterbirds and CelebA are standard benchmarks for spurious correlations robustness and group robustness, where for each data point both core and spurious feature are present and clear. CXR and FMOW are more challenging datasets with real-world distribution shift. The groups in FMOW are defined by the regions which are not equally represented in train data, but the spurious features (region identity) are more subtle and not present in all images. Moreover, FMOW additionally contains a domain shift across time: the validation and

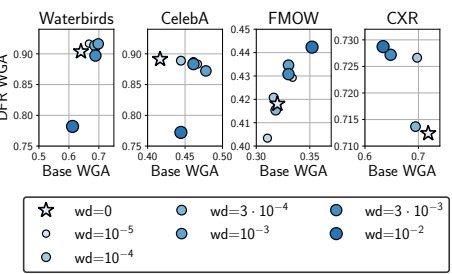

Figure 10: **Effect of weight decay.** While zero weight decay underperforms in base WGA on Waterbirds and CelebA, it provides near-optimal DFR WGA.

test images were taken after 2016, while the training images were taken before 2016. On CXR the spurious feature is only present for the positive class and completely absent for the negative class (see Section A.2 for detailed discussion). Consequently, we observe some differences in the results between the standard benchmarks and these more realistic datasets.

In Table 6 we report analogous results on the MultiNLI text classification problem.

**Effect of Data Augmentation.** Next, we consider 5 data augmentation policies: (1) no augmentations, (2) default augmentations, i.e. random crops and horizontal flips, (3) MixUp [98] combined with default augmentations, (4) Random Erasing [101], and (5) AugMix [29]. We train a ResNet-50 model pretrained on ImageNet1k on each of the four datasets with each of the augmentation policies, and report the results in Figure 9. We use the default models described in Appendix A.4, and default hyper-parameters in Table 2, and apply the data augmentation policies desribed in Appendix A.3.

AugMix provides the best performance on each dataset. However, we find that data augmentation is generally not required to achieve strong performance on any of the datasets with the exception of CXR: the model trained without augmentations is competitive across the board. Moreover, data augmentation can *hurt* the learned features. For example, while MixUp is helpful on Waterbirds and CelebA, it significantly hurts the preformance on FMOW, with 30% DFR WGA compared to 42% for the model trained without augmentation. We hypothesize that on the FMOW dataset, which contains highly detailed satellite images, mixing the images makes training overly challenging.

Similarly, Random Erasing hurts the DFR WGA on Waterbirds. We hypothesize that the randomly erased image block is more likely to fully cover the bird features than the background features, as the bird occupies a small fraction of the image relative to the background. Consequently, the model is incentivised to focus on the spurious feature: the model trained with Random Erasing learned the highest quality representation of the spurious feature across all augmentations with DFR $s$-WGA of $91.8 \pm 0.1\%$ across 3 runs, compared to e.g. $91.4 \pm 0.5\%$ for the model trained with no augmentation.

Balestriero et al. [5] report a related observation for models trained on ImageNet: in some cases data augmentation affects different classes disproportionately, and best data augmentation policies for mean accuracy can lead to poor worst-class accuracy,

**Effect of training length on ERM.** We plot the DFR WGA, DFR $s$-WGA, as well as base model WGA and mean accuracy as a function of training epoch for ERM training in Figure 11. On all datasets, we observe that the DFR WGA quickly converges and stays roughly constant throughout training. On all datasets, 5 epochs is sufficient for near-optimal performance. Longer training generally does not help or hurt DFR WGA, even when it hurts the base model.

**Effect of training length on Group-DRO.** We repeat the same experiment, but with Group-DRO instead of ERM training in Figure 12. Again, we observe that 5 epochs are generally sufficient for

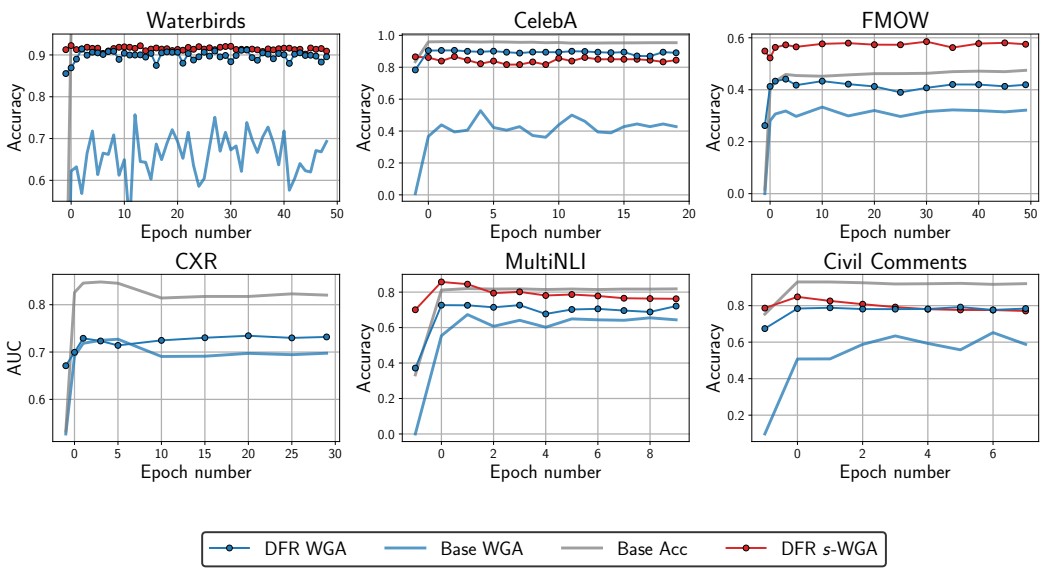

Figure 11: **ERM training length effect.** On all datasets, 5 epochs or less is sufficient to achieve near-optimal DFR WGA performance, but longer training does not hurt performance.

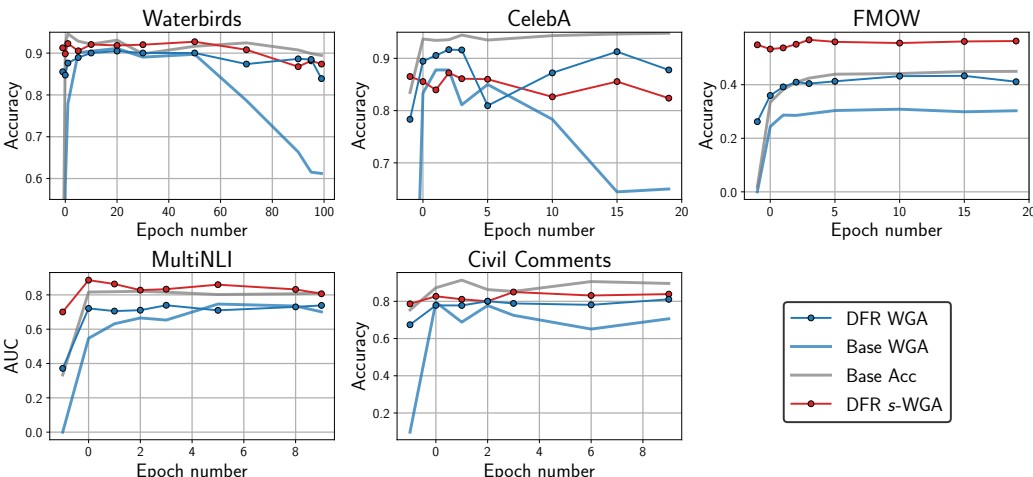

Figure 12: **Group-DRO training length effect.** DFR WGA converges to near-optimal performance in under 5 epochs. On Waterbirds and CelebA the DFR WGA performance is less stable compared to the ERM results in Figure 11.

near-optimal performance. Interestingly, for Group-DRO, DFR WGA does deteriorate over time on Waterbirds but not as significantly as the base model WGA.

# E  Additional results on MultiNLI

For all experiments in this section, we train the models for 5 epochs with learning rate $10^{-5}$ and 0 weight decay.

**Effect of base model.** In Table 3, we report the results for BERT-Base, BERT-Large, DeBERTa-Base, and DeBERTa-Large models [27, 28]. For BERT models, following Sagawa et al. [76], we use the cached tokenizer outputs with maximum sequence length 150. For DeBERTa models, we re-tokenize the dataset with the corresponding tokenizer with a maximum sequence length of

| Base Model | Base Acc | Base WGA | DFR WGA | DFR s-WGA |
|------------|----------|----------|---------|-----------|
| **BERT-Base** | 82.3 | 66.1 | 73.8 | 83.6 |
| **BERT-Large** | 84.6 | 70.6 | 76.6 | 79.0 |
| **DeBERTa-Base** | 88.9 | 80.4 | 82.3 | 80.2 |
| **DeBERTa-Large** | 90.2 | 81.2 | 84.8 | 68.3 |

Table 3: **MultiNLI: base model effect.** Effect of base model on the performance on the MultiNLI dataset. DeBERTa-Large model provides the best performance in terms of DFR WGA as well as Base WGA and Base mean accuracy.

| Pretraining | Base Acc | Base WGA | DFR WGA | DFR s-WGA |
|-------------|----------|----------|---------|-----------|
| **Random Init** | 59.1 | 26.0 | 45.4 | 94.9 |
| **Pretrained on Wiki + Book Corpus** | 82.3 | 66.1 | 73.8 | 83.6 |
| **MultiLingual on Wiki** | 80.5 | 62.1 | 72 | 83.9 |

Table 4: **MultiNLI: effect of pretraining.** Results for BERT-Base model with different types of pre-training on the MultiNLI dataset. Pretraining is required to achieve strong performance. Multilingual pretraining is competitive but inferior to pretraining on data in English.

220. We train all models for 5 epochs. The advanced DeBERTa-Large model provides the best base performance and DFR WGA.

**Effect of pretraining.** In Table 4 we evaluate the results of BERT-Base models with different types of pretraining on MultiNLI. This experiment is analogous to the experiment for image classification problems presented in Figure 5(a). We find that pretraining is necessary to achieve strong performance on this dataset, but different pretraining datasets lead to competitive results.

**Effect of training on target data.** In Table 5 we evaluate the effect of training on the MultiNLI dataset for the BERT and DeBERTa pretrained models. This experiment is analogous to the experiment for image classification problems presented in Figure 4. We find that after training on the target data both the core features (DFR WGA) and the spurious features (DFR $s$-WGA) become significantly more decodable. This result is in contrast to the results on Waterbirds in Figure 4, where DFR WGA is not significantly improved from training.

**Effect of weight decay.** In Table 6 we report the results of the weight decay ablation for the BERT-Base model on MultiNLI; this model uses the AdamW optimizer [53], so we consider larger values of weight decay.

# F   Broader Impact and Limitations

**Limitations.** While we consider a wide range of factors that affect the feature learning under spurious correlations, we inevitably do not cover all the possible factors. In particular, it would be interesting to consider the effect of regularization methods beyond weight decay and early stopping, and methods for diverse feature learning such as DivDis [48], or the method of Teney et al. [85]. As another limitation, while DFR performs well in our experiments, it is not guaranteed to learn an optimal linear classifier with the given features; further improvements in learning the last layer can be used to refine the results of our study. Despite these limitations, we believe that our work provides a comprehensive analysis of the feature learning under spurious correlations.

**Broader impact.** Research on spurious correlations is closely related to ML Fairness [16, 23, 41, 69, 1, 39]. We hope that our work can motivate further research in fairness, where techniques similar to DFR can be considered to improve the fairness of ML models. A potential negative outcome that can result from *misinterpretation* of our analysis is if the practitioners assume that spurious correlations are not an important issue, as ERM learns high quality representation of the core features. We emphasize that ERM still performs suboptimally (see Figure 1), as it does not provide a correct weighting for the features in the final classification layer. Spurious correlations are a significant practical issue that should be considered carefully in real-world applications.

| Pretraining | Init DFR WGA | Trained DFR WGA | Init DFR s-WGA | Trained DFR s-WGA |
|---|---|---|---|---|
| **BERT-Base** | 37.4 | 73.8 | 69.9 | 83.6 |
| **BERT-Large** | 34.4 | 76.6 | 57.6 | 81 |
| **DeBERTa-Base** | 58.5 | 82.3 | 77 | 80 |
| **DeBERTa-Large** | 61.8 | 84.8 | 66.4 | 68.3 |

Table 5: **MultiNLI: effect of training on target data.** Results for different models before and after training on the MultiNLI dataset. For all the considered models, both the core and the spurious features are significantly more decodabe after training on the target data.

| Weight decay | Base Acc | Base WGA | DFR WGA | DFR s-WGA |
|---|---|---|---|---|
| **0** | 82.3 | 66.1 | 73.8 | 83.6 |
| **1** | 81.4 | 63.2 | 74.6 | 90.1 |
| **3** | 74.5 | 43.0 | 66.0 | 94.4 |
| **10** | 63.5 | 23.4 | 39.9 | 66.0 |
| **30** | 57.0 | 15.2 | 28.9 | 59.3 |
| **100** | 46.4 | 1.7 | 3.2 | 53.0 |

Table 6: **MultiNLI: weight decay.** The effect of weight decay on the BERT-Base model on MultiNLI. Similarly to the results in Figure 10, weight decay 0 provides competitive performance. The best DFR WGA is achieved with weight decay 1.

**Compute.** We estimate the total compute used in the process of working on this paper at roughly 3000 GPU hours. The compute usage is dominated by the experiments presented in Figure 3, where we trained a large number of large-scale models on 4 vision datasets. The tuning of Group-DRO hyper-parameters was also relatively compute-heavy. The experiments were run on GPU clusters on Nvidia Tesla V100, Titan RTX, RTX8000, 3080 and 1080Ti GPUs.

**Licenses.** The Civil Comments dataset is distributed under the CC0 license. The FMOW dataset is under the FMoW Challenge Public License. The Places dataset is under the CC BY license. For the details of the license for the MultiNLI dataset, see Williams et al. [91].