# OpenReview forum: "On Feature Learning in the Presence of Spurious Correlations"
_NeurIPS.cc/2022/Conference — NeurIPS 2022 Accept_

### Official Review · Reviewer_DoXQ · 2022-06-24

**Rating:** 7
**Confidence:** 4
**Soundness:** 3 good
**Presentation:** 4 excellent
**Contribution:** 3 good

**Summary:**

This is primarily an empirical paper about learning in the presence of spurious correlations. They run a lot of tests that show training a model with empirical risk minimization (ERM) followed by deep feature reweighting (DFR) (which is retraining last layer on some on a held-out set that doesn't have spurious correlations) yields results that are not too different from group robustness training (Group DRO)

**Questions:**

My questions are all integrated into "Strengths and Weaknesses"

**Limitations:**

Not required here.

**Strengths And Weaknesses:**

Thanks to the authors for the hard work on this paper. I like it overall, and think it is a valuable contribution.

Strengths
------------
- A large amount of insightful experiments
- Clear writing
- Sensible comparisons and conclusions

Weaknesses
----------------
- The conclusion of this paper hinges on the idea that: "If two different methods get roughly the same held-out performance, then they must be learning the same kind of thing." I don't think that's necessarily true, and your experiments don't really prove it. The conclusion in lines 207-211 could be made stronger with other types of analyses. To have more certainty it's that "better weighting of the learned features rather than learning better representations of the core features" you could, for example, actually extract the representations learned from ERM+DFT and GDRO and see if they look similar. This experiment could be tricky, but if you make sure to have identical weight initialization, and run the experiment ~10 times, you could show that the feature spaces learned are really similar or not.
- Why is early stopping important for RWY, RWG and GDRO? Are there some simple experiments you could do to elucidate this?

---

> ### Author Response · Authors · 2022-08-02
> **Thank you for your review!**
>
> Thank you for your positive review!
>
> ## New experiments on comparing feature representations
>
> Based on your feedback, we performed new experiments comparing the feature representations learned by GroupDRO and ERM on the Waterbirds and CelebA datasets using the UMAP dimensionality reduction. We report the results in Appendix F, Figure 12. The network weights were initialized to the same values for both methods.
>
> Generally, as you mention, it is hard to make strong conclusions about the similarity of the learned representations. In our work (in particular, in lines 207-211) we do **not** argue that Group DRO and ERM learn the same  feature representations, but instead that **the core features are equally decodable from the representations learned by these methods**. This is why we consider the DFR WGA as the main metric in the paper: it measures how well we can decode the core features with a linear probe. Our results show that even if the exact form of the learned representations is different, the amount of information about the core features in the representations is similar between different methods.
>
> ## Early stopping
>
> We first note that while early stopping is helpful for RWY, RWG and GDRO Base WGA performance, it does not significantly improve DFR WGA for these methods. Consequently, we conclude that early stopping is a useful regularization technique for learning a better weighting of the features (in the last layer), but does not necessarily allow the model to learn more information about the core features. Why the Base WGA performance of  RWY, RWG and GDRO deteriorates so significantly with training time is an interesting open question for future work.
>
> **Inspired by your feedback, we added new experiments on comparing the feature representations learned by ERM and Group DRO.** Please feel welcome to ask any further questions, and we will respond if the system permits.

---

> > ### Comment · Reviewer_DoXQ · 2022-08-08
> > **Thank you.**
> >
> > Thank you for your response and the additional new content.

---

### Official Review · Reviewer_CCN4 · 2022-07-11

**Rating:** 6
**Confidence:** 4
**Soundness:** 4 excellent
**Presentation:** 3 good
**Contribution:** 3 good

**Summary:**

This paper considers deep learning in the common case where the training data contains spurious correlations. The main takeaway is that empirical risk minimization (alone) is sufficient to obtain state-of-the-art performance; specialized group robustness methods do not appear to provide a significant benefit. This is demonstrated on six datasets spanning both vision and text problems. The effect of the architecture, pretraining strategy, and regularization is also considered.

**Questions:**

For all datasets considered, I assume the labeled spurious attributes form an *incomplete* labeling (there always exists some latent spurious attributes). How do latent spurious features impact the conclusions of the work?

**Limitations:**

I would have liked to see the scope of the paper defined a bit more clearly. In addition, while both vision and text datasets are used, the majority of the experiments are only on the image datasets.

**Strengths And Weaknesses:**

Spurious correlations are a concern when fitting neural networks, so this paper tackles an important problem. Overall, I found the presentation to be quite good and the experiments fairly convincing. My main issue with the work in its current form is that the scope (and therefore potential impact) of the work is more limited than the title and introduction suggest: the spurious correlations studied are *labeled* properties of the inputs, rather than latent spurious features. In most cases, one does not have access to labeled attributes (or even class labels!) when fitting a neural network, and therefore this work has a narrower scope than expected.

This being the case, I think it is notable that specialized group robustness methods appear to perform no better than ERM when it comes to learning the in the presence of spurious correlations. In addition, the empirical observations regarding regularization and other effects of the base model are interesting, although many of them rely on DFR, which AFAIK is not peer reviewed.

The DFR procedure is somewhat similar to what's done in contrastive learning, e.g. "Supervised Contrastive Learning," except there the second stage is performed on the original dataset. This often results in improved performance thanks to the contrastive objective. I'm curious how supervised contrastive learning would impact the results, both using the original dataset or the "reweighted" one.

I would have liked to see the analysis of pretraining to include additional experiments with text / BERT.

A smaller concern is that this paper leans heavily on Deep Feature Reweighting (DFR), which appears in a recent arxiv preprint (Kirichenko et al. [22]). Unfortunately, reading that pre-print is necessary to understand this work; reading the short description in S3 was not sufficient to follow along. It would be better to make this paper self-contained, especially given how simple the DFR idea is (e.g., define the reweighting dataset).

---

> ### Author Response · Authors · 2022-08-02
> **Thank you for your review!**
>
> Thank you for your thoughtful feedback!
>
> ## Spurious labels and the scope of the paper.
>
> We note that our paper is primarily an analysis paper: we are studying how well neural networks represent the core features when the data contains spurious features. It is true that in all of our experiments the spurious features are known and labeled, but the results of the analysis will apply to cases when the spurious features are unknown as well!
>
> For example, ERM does not use the spurious label information in any way, so the features that are learned by ERM are unaffected by whether or not the spurious labels are known to us. Our observations about the quality of the core feature representations learned by ERM will hold in the cases when the spurious features are unknown.
>
> To sum up, all of the main conclusions of the paper (ERM vs group robustness training, effect of pretraining, effect of data augmentation, effect of regularization, …) will hold in practice when the spurious attributes are not known to us, as our analysis does not rely on the spurious labels being known.
>
> We added a clarification on this point in lines 116-121. We will clarify this point further.
>
> ## New experiments on text models.
>
>
> Based on your suggestions, we performed several additional experiments with the text models. First, as you requested, we performed an experiment on the effect of pretraining on the MultiNLI dataset with the BERT-Base model [BERT-Base](https://huggingface.co/bert-base-uncased) model architecture. We compare three parameter initializations: random initialization, [model pretrained on Wikipedia and Book Corpus](https://huggingface.co/bert-base-uncased) and [multilingual model pretrained on Wikipedia](https://huggingface.co/bert-base-multilingual-uncased).
>
> |                                  | Base Acc | Base WGA | DFR WGA | DFR s-WGA |
> | -------------------------------- | -------- | -------- | ------- | --------- |
> | Random Init                      | 59.1     | 26.0     | 45.4    | 94.9      |
> | Pretrained on Wiki + Book Corpus | 82.3     | 66.1     | 73.8    | 83.6      |
> | MultiLingual on Wiki             | 80.5     | 62.1     | 72.     | 83.9      |
>
> Similar to the image classification results, we find that pretraining is required to achieve strong performance on this problem. Multilingual pretraining provides competitive but inferior performance to the standard pretraining. The MultiNLI dataset contains sentences in English, so targeted pretraining on data in English provides the best results.
>
> Inspired by your comments, we also repeated the other experiments from the paper on the MultiNLI dataset: we evaluate the effects of the model architecture, training on the target data, regularization strength, and length of training. Please see our [general response](https://openreview.net/forum?id=wKhUPzqVap6&noteId=3imPpVvUQ1P) for detailed results.
>
> We hope that you will consider these new results in your final evaluation of our paper.
>
> ## Other comments
>
> **Connections to contrastive learning.**
> Thank you for your suggestion to consider contrastive learning in terms of robustness to spurious correlations. We note that in the paper we already consider the effect of contrastive pretraining on ImageNet on the quality of learned representations on problems with spurious correlations. The results are presented in Figure 5: contrastive methods generally provide similar results to supervised pretraining.
>
> Contrastive supervised learning to reduce the effect of spurious correlations has also recently been considered in [1], where it was shown that the contrastive loss is generally more robust to noise in the spurious attribute labels, but does not provide significantly better results than group DRO when the correct spurious attribute labels are available.
>
> **Reliance on Kirichenko et al., 2022.**
> We updated the description on DFR in Section 3 to make the presentation more self-contained (lines 128-133). While it is true that our methodology is based on [2], the DFR method in [2] is very simple, as you mention. Moreover, we believe that the results in our paper are self-sufficient, and do not rely on the correctness of observations in [2].
>
> **Based on your feedback, we added several new experiments on text models, clarified the scope of the paper and updated the discussion of the DFR method.** Given these updates, we hope that you will consider raising your score. Please feel welcome to ask any further questions, and we will respond if the openreview system permits.
>
> **References**
>
> [1] ​​*Correct-N-Contrast: a Contrastive Approach for Improving Robustness to Spurious Correlations*;
> Michael Zhang, Nimit S Sohoni, Hongyang R Zhang, Chelsea Finn, Christopher Re
>
> [2] *Last Layer Re-Training is Sufficient for Robustness to Spurious Correlations*;
> Polina Kirichenko, Pavel Izmailov, Andrew Gordon Wilson

---

> > ### Comment · Reviewer_CCN4 · 2022-08-08
> > **Thanks!**
> >
> > Thanks for the response! I have revised my recommendation in light of this new information.

---

### Official Review · Reviewer_amrA · 2022-07-11

**Rating:** 7
**Confidence:** 4
**Soundness:** 3 good
**Presentation:** 4 excellent
**Contribution:** 3 good

**Summary:**

This paper studies the reliance of deep learning models on spurious correlations. In particular, the authors look at the quality of feature representations learned by models trained via ERM versus models trained using group robustness methods. They evaluate these feature representations by utilizing the Deep Feature Retraining (DFR) procedure: retraining the last layer of the model on a held-out set which would likely not contain spurious correlations present in the training set. This procedure helps reveal how much information about causal factors is present in the learned representations. The authors further explore how these feature representations are influenced by the model architecture, pre-training task and strategy, regularization (via weight decay, data augmentation), training length, and whether or not model has been trained on the target data. They find that the quality of these representations often depends heavily on choice of data augmentation, model architecture, and the pre-training strategy involved, while regularization and training time may not be as helpful in improving the quality of said representations. The authors share results on CelebA, Waterbirds, WILDS-FMOW, CXR, MultiNLI, and CivilComments datasets.

Overall, the paper presents interesting results and insights. I have some minor comments which I hope the authors will address during the response period.

**Questions:**

- Line 364: What changes in each of these three runs?
- Regarding results on CXR: You suggest that the lack of improvement is because the model already learns the features correctly and DFR is unable to improve much upon it. Could it also be the case that this is one of the cases of failure modes for DFR?
- Regarding effect of regularization on FMOW: Is it plausible that the trends are this way because models trained on FMOW are already regularized quite significantly (given the amount of noise in the images), and increasing regularization (through weight decay or MixUp) is weakening the signal further?

**Limitations:**

As this is an analysis paper, it is hard to understand the limitations and potential negative social impact, but I would urge the authors to think about potential negative impacts arising from misinterpretation of their analysis.

**Strengths And Weaknesses:**

### Strengths:
- This work looks at an interesting and well motivated problem.
- The experimental setup is well designed and results offer insights that would be useful to future researchers.
- The paper is well written and organized.

### Weaknesses:
- The experiments on NLP datasets are based on a BERT model. While I understand that the goal here is not to create a state of the art model but to analyze representations learned by a model, significantly better models (DeBERTa, ERNIE, T5, etc) are out there that the authors could have used.
- There are several works in NLP that have looked at the problem of spurious correlations ([1,2,3,4] are just a few examples), addressing them and understanding when models weigh causal features vs non causal features. The paper currently does not position itself well in that literature.

### Additional comments:
- Section 3, Preliminaries (Lines 99-102): This appears to be incorrect in light of [1]. In fact, as most Machine Learning tasks are anticausal, models will rely on spurious correlations regardless. As [1] show in their anticausal setup, a model will rely on spurious factors most of the time (unless the spurious features observe higher noise compared to the causal features).

[1] Kaushik, D., Setlur, A., Hovy, E. H., & Lipton, Z. C. Explaining the Efficacy of Counterfactually Augmented Data. ICLR 2021.
[2] Eisenstein, J. (2022). Uninformative Input Features and Counterfactual Invariance: Two Perspectives on Spurious Correlations in Natural Language. arXiv preprint arXiv:2204.04487.
[3] Veitch, V., D'Amour, A., Yadlowsky, S., & Eisenstein, J. Counterfactual invariance to spurious correlations in text classification. NeurIPS 2021.
[4] Kaushik, D., Hovy, E., & Lipton, Z. Learning The Difference That Makes A Difference With Counterfactually-Augmented Data. ICLR 2020.

---

> ### Author Response · Authors · 2022-08-02
> **Thank you for your review!**
>
> Thank you for your supportive review!
>
> ## New experiments on text models.
> Inspired by your feedback we conducted new experiments on text problems with different model architectures. In particular, in addition to the BERT-Base model, we experimented with DeBERTa-Base, DeBERTa-Large and BERT-Large on the the MultiNLI dataset. We chose DeBERTa because it has a convenient official implementation in the huggingface transformers package. With these models, we achieved the following results:
>
> |               | Base Acc | Base WGA | DFR WGA | DFR s-WGA |
> | ------------- | -------- | -------- | ------- | --------- |
> | BERT-Base     | 82.3     | 66.1     | 73.8    | 83.6      |
> | BERT-Large    | 84.6     | 70.6     | 76.6    | 79.0      |
> | DeBERTa-Base  | 88.9     | 80.4     | 82.3    | 80.2      |
> | DeBERTa-Large | 90.2     | 81.2     | 84.8    | 68.3      |
>
> As expected, the DeBERTa models provide improved performance across the board, with better base model mean accuracy, base WGA and DFR WGA. BERT-Large and DeBERTa-Large models outperform the respective Base models. Interestingly, BERT-Base learns the most informative representations for predicting the spurious feature, as measured by DFR s-WGA; large models store less information about the spurious feature in the learned representations.
>
> In addition to these experiments, we extended all of the applicable analysis experiments in the paper to the MultiNLI dataset: we evaluate the effects of training on the target data, pretraining datasets, regularization strength, and length of training. Please see our [general response](https://openreview.net/forum?id=wKhUPzqVap6&noteId=3imPpVvUQ1P) for detailed results.
>
> ## Literature on spurious correlations in NLP.
>
> Thank you for these relevant pointers! We updated the paper to include a discussion of these works, see lines (lines 82-87).
>
> ## Other Questions
>
> *Lines 99-102 (Preliminaries): models will rely on the spurious features regardless?*
>
> In those lines we are simply stating that if a feature $s$ is correlated with $y$ on the training data, the model may rely on the attribute $s$ to make predictions about $y$. If $s$ and $y$ are not correlated in the training data, the model is not incentivised to rely on $s$ in its predictions, and in this case $s$ is not a spurious feature.
>
> *Line 364: What changes in each of these three runs?*
>
> The three runs used different random seeds. We held all other parameters fixed.
>
> *Could it be that CXR is one of the cases of failure modes for DFR?*
>
> CXR is indeed a dataset where DFR fails to significantly improve performance compared to the base model. We believe that the reason behind this result is that on CXR the spurious feature never hurts predictions: all patients with a chest drain are labeled as sick. Consequently, even if DFR was able to learn a classifier that ignores the chest drain, this classifier would not have improved the worst group accuracy. In contrast, on a dataset like Waterbirds, removing the reliance on the background feature improves the performance on the mixed groups, as there are test examples where the background contradicts the foreground type. We discuss this point in more detail in the appendix A.2 of the updated paper.
>
> *Is it possible that FMOW is sufficiently regularized by the noise in the inputs, and that is why weight decay and mixup hurt?*
>
> Note that according to Figure 6, FMOW is in fact the dataset where weight decay is the most useful! Data augmentations other than MixUP are not particularly helpful, but also do not hurt the results significantly. We believe that your interpretation is correct for MixUp: the data already is noisy and contains fine detail, and mixing the images with MixUp seems to hurt the performance.
>
> *I would urge the authors to think about potential negative impacts arising from misinterpretation of their analysis.*
>
> We added a discussion of a danger of potential misinterpretation of our analysis to the “Broader Impact” section (lines 912-917):
>
> > A potential negative outcome that can result from misinterpretation of our analysis is if the practitioners assume that spurious correlations are not an important issue, as ERM learns high quality representation of the core features. We emphasize that ERM still performs suboptimally (see Figure 1), as it does not provide a correct weighting for the features in the final classification layer. Spurious correlations are a significant practical issue that should be considered carefully in real-world applications.
>
>
> **Based on your comments, we updated the paper with the new experimental results on the text data, and extended the discussion of related work.** We hope this response helps address your question. Please feel welcome to ask us if you have any further questions, and we will respond if the system permits.

---

> > ### Comment · Reviewer_amrA · 2022-08-07
> > **Thanks for the response**
> >
> > Appreciate the additional experiments using DeBERTa, and the additions/clarification to your paper as offered in the updated version. I feel confident accepting this paper.
> >
> > One minor comment: There seems to be a new line instead of a space after the comma on Line 99.

---

### Author Response · Authors · 2022-08-02
**General Response (part 1)**

We thank the reviewers for thoughtful and supportive feedback. We want to highlight that the paper is making many timely and significant observations about the quality of learned feature representations in the presence of spurious correlations:
- We show that the quality of core feature representations learned by ERM is comparable to that of state-of-the-art group robustness methods such as Group DRO. The improvements in performance of group robustness methods can largely be explained by learning a better classification last layer, rather than learning better features.
- We show that the model architecture and pretraining have a significant impact on the quality of feature representations. The worst group accuracy with an optimal linear last layer obtained by DFR is close to linearly correlated with the base model average accuracy: models with better in-distribution performance also learn better features for minority groups.
- We show that it is possible to improve upon the best reported performance in the literature on Waterbirds and CelebA with simple ERM training followed by DFR.
- We show that on the popular Waterbirds dataset, it is possible to achieve state-of-the-art performance without training on the dataset at all. This observation shows that the Waterbirds dataset should not be used as the main dataset to evaluate group robustness methods, and especially the quality of learned feature representations, as was often done in prior work.
- We provide detailed results on the effect of data augmentation, pretraining, regularization and training length, which will be of interest to both researchers and practitioners working on spurious correlations.

## New experiments

Based on the feedback from the reviewers we performed multiple new experiments and updated the paper.

### Results on NLP problems

Based on the requests from reviewers, we significantly expand our exploration on the NLP problems. The results in all experiments agree with our observations on the vision problems. We summarize the results on the MultiNLI dataset below.

**Advanced architectures.**
First, we evaluate three new architectures on the MultiNLI dataset: [BERT-Large](https://huggingface.co/bert-large-uncased), [DeBERTa-Base](https://huggingface.co/microsoft/mdeberta-v3-base) and [DeBERTa-Large](https://huggingface.co/microsoft/deberta-v3-large); we also report the results for the [BERT-Base](https://huggingface.co/bert-base-uncased) which was used throughout our original submission.
With these models, we achieve the following results:

|               | Base Acc | Base WGA | DFR WGA | DFR s-WGA |
| ------------- | -------- | -------- | ------- | --------- |
| BERT-Base     | 82.3     | 66.1     | 73.8    | 83.6      |
| BERT-Large    | 84.6     | 70.6     | 76.6    | 79.0      |
| DeBERTa-Base  | 88.9     | 80.4     | 82.3    | 80.2      |
| DeBERTa-Large | 90.2     | 81.2     | 84.8    | 68.3      |

As expected, DeBERTa-Large provides the best in-distribution accuracy, WGA, and DFR-WGA. Interestingly, DFR $s$-WGA is the lowest for this model, indicating that the feature representations that it learns contain the smallest amount of information about the spurious feature.

**Effect of training on the target data.** We perform the experiment on the effect of training on the target data, analogous to the image classification experiment in Figure 4, on the MultiNLI dataset. For each model and dataset we evaluate DFR WGA and DFR s-WGA for the model before and after training on the target dataset.

|               | Init DFR WGA | Trained DFR WGA | Init DFR s-WGA | Trained DFR s-WGA |
| ------------- | ------------ | --------------- | -------------- | ----------------- |
| BERT-Base     | 37.4         | 73.8            | 69.9           | 83.6              |
| BERT-Large    | 34.4         | 76.6            | 57.6           | 81                |
| DeBERTa-Base  | 58.5         | 82.3            | 77             | 80                |
| DeBERTa-Large | 61.8         | 84.8            | 66.4           | 68.3              |

Training on the target dataset is necessary to achieve strong performance on this dataset with all models. Both DFR WGA and DFR $s$-WGA are improved after training.

---

> ### Author Response · Authors · 2022-08-02
> **General Response (part 2)**
>
> **Effect of pretraining.** We perform the experiment on the effect of pre-training, analogous to the image classification experiment in Figure 5 (a), on the MultiNLI dataset with the [BERT-Base](https://huggingface.co/bert-base-uncased) model architecture. We compare three parameter initializations: random initialization, [model pretrained on Wikipedia and Book Corpus](https://huggingface.co/bert-base-uncased) and [multilingual model pretrained on Wikipedia](https://huggingface.co/bert-base-multilingual-uncased). We note that the multilingual model uses a different tokenizer compared to the other two models.
>
> |                                  | Base Acc | Base WGA | DFR WGA | DFR s-WGA |
> | -------------------------------- | -------- | -------- | ------- | --------- |
> | Random Init                      | 59.1     | 26.0     | 45.4    | 94.9      |
> | Pretrained on Wiki + Book Corpus | 82.3     | 66.1     | 73.8    | 83.6      |
> | MultiLingual on Wiki             | 80.5     | 62.1     | 72.     | 83.9      |
>
> Similar to the vision problems, pretraining is helpful for achieving strong results on MultiNLI. In particular, the [model pretrained on Wikipedia and Book Corpus](https://huggingface.co/bert-base-uncased) achieves the best performance across the considered pretraining strategies.
>
> **Effect of training length.** The results on the effect of training length are reported in the Appendix Figure 11 for all datasets, both vision and NLP. We find that long training is not necessary to achieve near-optimal DFR WGA performance.
>
> **Effect of weight decay.** We repeat the experiment on the effect of weight decay, analogous to the image classification experiment in Figure 6, on the MultiNLI dataset with the [BERT-Base](https://huggingface.co/bert-base-uncased) model. Note that the scale of weight decay values is much larger for the text problems compared to vision problems, as these models use the AdamW optimizer instead of SGD.
>
> | Weight decay | Base Acc | Base WGA | DFR WGA | DFR s-WGA |
> | ------------ | -------- | -------- | ------- | --------- |
> | 0.           | 82.3     | 66.1     | 73.8    | 83.6      |
> | 1.           | 81.4     | 63.2     | 74.6    | 90.1      |
> | 3.           | 74.5     | 43.0     | 66.0    | 94.4      |
> | 10.          | 63.5     | 23.4     | 39.9    | 66.0      |
> | 30.          | 57.0     | 15.2     | 28.9    | 59.3      |
> | 100.         | 46.4     | 1.7      | 3.2     | 53.0      |
>
> Similar to the results on the image classification problems, zero weight decay provides competitive performance, while high weight decay values lead to poor results, both in terms of Base WGA and DFR WGA.
>
>
> ### Learned feature visualizations
>
> Following the suggestion of reviewer DoXQ we add feature visualizations for ERM and Group DRO in the Appendix F. We discuss the results in detail in the response to reviewer DoXQ.
>
>
> ## Updates to the paper
>
> Here we list the updates we made to the paper to address the questions and suggestions raised by the reviewers. We uploaded a new version of the paper, where we highlight the updated parts of the text in blue. In particular, we
> - Extend the discussion of related work in NLP (lines 82-87).
> - Clarify the problem setting and the need for spurious feature labels (lines 102-120).
> - Clarify the DFR procedure and the definition of reweighting dataset (lines 127-132).
> - Add new results on the MultiNLI dataset to the appendix (tables 3-6, discussion in Appendices C, D, E)
> - Add the experiment on feature visualizations to the appendix (Figure 12, Appendix F).
> - Expand the discussion of broader impact (lines 912-917).
>
> In order to keep the main text within the page limit, we made minor edits and temporarily moved the “Are VITs more robust than CNNs?” paragraph to the appendix.  We will move it back to the main text for the camera ready, if the page limit permits.
>
> We now respond to each reviewer individually.

---

### Meta-Review · Area_Chair_3YMu · 2022-08-28

**Recommendation:** Accept
**Confidence:** Certain

**Metareview:**

The paper shows that empirical risk minimization is sufficient to obtain good worst-group accuracies and specialized group robustness methods do not appear to provide additional benefits.

The reviewers pointed out that the current work depends on DFR, which seems to require some additional data compared to group robustness methods. The reviewers also note that the NLP experiments did not use more recent models, and the authors addressed these issues. Generally the reviewers think this is a well-executed paper on an important problem, and are unanimous in accepting it.

**Award:**

No

---

### Decision · Program_Chairs · 2022-09-14

Accept